



**Understanding Processes that Control Dust Spatial Distributions with**
**Global Climate Models and Satellite Observations**
Mingxuan Wu[1,2], Xiaohong Liu[1,3], Hongbin Yu[4], Hailong Wang[2], Yang Shi[1,3], Kang Yang[5], Anton Darmenov[4],
Chenglai Wu[1], Zhien Wang[5], Tao Luo[1], Yan Feng[6], Ziming Ke[1,3]
[1]Department of Atmospheric Science, University of Wyoming, Laramie, WY, USA
[2]Atmospheric Sciences and Global Change Division, Pacific Northwest National Laboratory, Richland, WA,
USA
[3]Department of Atmospheric Sciences, Texas A&M University, College Station, Texas, USA
[4]NASA Goddard Space Flight Center, Greenbelt, Maryland, USA
[5]Department of Atmospheric and Oceanic Sciences, University of Colorado Boulder, Boulder, CO, USA
[6]Environmental Science Division, Argonne National Laboratory, Argonne, IL, USA
*Correspondence to*: Xiaohong Liu (xiaohong.liu@tamu.edu)
**Abstract**
Dust aerosol is important in modulating the climate system at local and global scales, yet its spatiotemporal
distributions simulated by global climate models (GCMs) are highly uncertain. In this study, we evaluate the
spatiotemporal variations of dust extinction profiles and dust optical depth (DOD) simulated by the
Community Earth System Model version 1 (CESM1) and version 2 (CESM2), the Energy Exascale Earth
System Model version 1 (E3SMv1), and the Modern-Era Retrospective analysis for Research and
Applications version 2 (MERRA-2) against satellite retrievals from Cloud-Aerosol Lidar with Orthogonal
Polarization (CALIOP), Moderate Resolution Imaging Spectroradiometer (MODIS), and Multi-angle
Imaging SpectroRadiometer (MISR). We find that CESM1, CESM2, and E3SMv1 underestimate dust
transport to remote regions. E3SMv1 performs better than CESM1 and CESM2 in simulating dust transport
and the northern hemispheric DOD due to its higher mass fraction of fine dust. CESM2 performs the worst
in the northern hemisphere due to its lower dust emission than in the other two models but has a better dust
simulation over the Southern Ocean due to the overestimation of dust emission in the southern hemisphere.
DOD from MERRA-2 agrees well with CALIOP DOD in remote regions due to its higher mass fraction of





fine dust and the assimilation of aerosol optical depth. The large disagreements in the dust extinction profiles
and DOD among CALIOP, MODIS, and MISR retrievals make the model evaluation of dust spatial
distributions challenging. Our study indicates the importance of representing dust emission, dry/wet
deposition, and size distribution in GCMs in correctly simulating dust spatiotemporal distributions.

## 1 Introduction

Mineral dust plays an important role in the Earth's climate system. It can impact the Earth's radiation
budget directly through scattering and absorbing solar and terrestrial radiation (e.g., Tegen et al., 1996;
Balkanski et al., 2007), and indirectly through acting as cloud condensation nuclei and ice nucleating particles
(e.g., Rosenfeld et al., 2001; DeMott et al., 2003; Shi and Liu, 2019). Dust can reduce the snow albedo when
deposited on snow (e.g., Yasunari et al., 2015; Wu et al., 2018b; Rahimi et al., 2019), participate in the
heterogeneous atmospheric chemistry reactions (e.g., Dentener et al., 1996), and provide nutrients such as
iron to oceans through deposition (e.g., Jickells et al., 2005). Dust aerosols are reported to have a negative
radiative forcing (RF) due to aerosol-radiation interactions (RFari); however, large uncertainties exist in the
dust RFari estimates (Boucher et al., 2013). Whether mineral dust warms or cools the climate is still
controversial (e.g., Boucher et al., 2013; Scanza et al., 2015; Kok et al., 2017).
The large uncertainties in estimating dust RFari can be mainly attributed to the large diversities in the
dust lifecycle (i.e., emission, transport and deposition) simulated by current global climate models (GCMs)
(e.g., Huneeus et al., 2011; Boucher et al., 2013; Kim et al., 2014, 2019; Pu & Ginoux, 2018; Wu et al.,
2018a), which is not well constrained by observations. Huneeus et al. (2011) found that global total dust
emission from 14 GCMs participating in the Aerosol Comparisons between Observations and Models
(AeroCom) Phase I ranges from 514 to 4313 Tg yr$^{-1}$ while global annual mean dust optical depth (DOD)



ranges from 0.010 to 0.053. Pu and Ginoux (2018) showed that the Coupled Model Intercomparison Project
Phase 5 (CMIP5) models underestimate DOD, especially in spring, compared with land DOD derived from
MODIS. Wu et al. (2018a) found that dust emission from CMIP5 models differs greatly in spatial distribution
and intensity over East Asia. Kim et al. (2014, 2019) compared DOD from 5 GCMs participating in the
AeroCom Phase II with DOD derived from the Cloud-Aerosol Lidar with Orthogonal Polarization (CALIOP),
Moderate Resolution Imaging Spectroradiometer (MODIS), and Multiangle Imaging Spectroradiometer
(MISR) in the trans-Atlantic and trans-Pacific regions, respectively. Large diversities are found in the
modeled DOD over the source regions of North Africa and East Asia, implying large uncertainties associated
with dust emissions in these models. The low model biases of DOD across the North Atlantic and North
Pacific indicate that current GCMs underestimate the trans-Atlantic transport of North African dust and the
trans-Pacific transport of East Asian dust, respectively, likely due to an overestimation of dust removal.

Apart from horizontal distribution, the vertical distribution of mineral dust can strongly influence the

radiative effects of dust (e.g., Zhang et al., 2013), which is poorly constrained by observations. Few studies
directly compared dust extinction profiles in GCMs with retrievals from CALIOP onboard Cloud-Aerosol
Lidar and Infrared Pathfinder Satellite Observation (CALIPSO) (e.g., Yu et al. 2010; Johnson et al., 2012;
Kim et al., 2019; Wu et al., 2019). Yu et al. (2010) separated the dust extinction from the total aerosol
extinction in the nighttime cloud-free CALIOP level 2 (CAL-L2) version 2.01 product using the volume
depolarization ratio. They compared the dust extinction simulated by the Goddard Chemistry Aerosol
Radiation Transport (GOCART) model with CALIPSO observations from June 2006 to November 2007.
Johnson et al. (2012) evaluated the dust extinction simulated by GEOS-Chem, a global 3-D chemical
transport model driven by meteorological input from the Goddard Earth Observing System (GEOS), with
CAL-L2 version 3.01 product from March 2009 to February 2010 and found high model biases of dust


extinction in the lower troposphere over main source regions, similar as Yu et al. (2010). Wu et al. (2019)
compared dust extinction modeled by the Community Earth System Model (CESM) with satellite retrievals
from Luo et al. (2015a, 2015b) (L15), Yu et al. (2015) (Y15), and standard CALIOP level 3 (CAL-L3) product
and found high model biases of dust extinction in the upper troposphere and large uncertainties in different
CALIPSO products over East Asia.
A major challenge in evaluating mineral dust in GCMs is the lack of high-quality and long-term
measurements of dust (Evan et al., 2014). The limited spatiotemporal coverage of ground-based and aircraft
observations is insufficient to provide global scale dust information. Pu and Ginoux (2016) derived DOD
over land from MODIS Deep Blue aerosol products using Ångström exponent and single scattering albedo.
Compared to coarse mode aerosol optical depth (AOD) from Aerosol Robotic Network (AERONET) ground-
based observations, MODIS DOD over land is slightly underestimated. Yu et al. (2009) derived DOD over
ocean from MODIS Dark Target aerosol products using prescribed fine mode fractions of combustion, dust,
and marine aerosols. MODIS DOD over ocean shows that Asian dust can contribute substantially to the
aerosol loading over North America (Yu et al., 2012). Luo et al. (2015a) developed a dust separation method
to retrieve dust extinction from CAL-L1B product, which gives lower dust extinction in the lower troposphere
(< 4 km) than CAL-L2 product. Luo et al. (2015b) developed a dust identification method to better detect
optically thin dust layers and found significantly frequent dust occurrences in the upper troposphere than
CAL-L2 product. Ridley et al. (2016) estimated the global DOD to be 0.030 ± 0.005 by combining satellite
retrievals of AOD with DOD simulated by four global models, which is close to AeroCom mean (0.028 ±
0.011, Huneeus et al., 2011) but has less uncertainties.
In this study, we compare dust extinction profiles and DOD simulated from CESM1, CESM2, the Energy
Exascale Earth System Model version 1 (E3SMv1) and the Modern-Era Retrospective analysis for Research



and Applications version 2 (MERRA-2) with satellite retrievals from CALIOP (L15 and Y15), MODIS, and
MISR on a global scale. We pay attention not only to the physical processes responsible for the model biases
of dust but also to the uncertainties in satellite retrievals and the impacts of these uncertainties on the model
evaluation. The goal of this study is to evaluate the performance of CESM1, CESM2, E3SMv1, and MERRA-
2 in the simulations of (1) dust mass budgets, (2) dust extinction profiles and DOD, and (3) dust surface
concentrations. The paper is organized as follows. Section 2 first introduces the models (CESM1, CESM2,
E3SMv1, and MERRA-2), and then gives a detailed description of the satellite retrievals used in this study.
Section 3 first shows the global dust mass budgets from the four models and then compares modeled dust
extinction profiles and DOD with satellite retrievals. Discussion and conclusions are presented in section 4.

**2 Models and Data**

In this section, we give a brief description of the GCMs (Section 2.1), experiments design (Section 2.2),

and satellite retrievals (Section 2.3) used in this study. Some important model features for simulating dust in
CESM1, CESM2, E3SMv1, and MERRA-2 are summarized in Table 1.

**2.1 Model Description**
**2.1.1 CESM**

In this study, we use the latest CESM2.1 with the Community Atmosphere Model version 6 (CAM6) and

the Community Land Model version 5 (CLM5, Lawrence et al., 2019) as the atmosphere and land component,
respectively. CAM6 has replaced earlier schemes for boundary layer turbulence, shallow convection and
cloud macrophysics with the Cloud Layers Unified by Binormals (CLUBB, Golaz et al., 2002; Bogenschutz
et al., 2013) scheme. CAM6 uses an improved two-moment cloud microphysics (MG2, Gettelman and


Morrison, 2015) scheme and the four-mode version of Modal Aerosol Module (MAM4, Liu et al., 2016).
Dust is represented in the Aitken mode, accumulation mode, and coarse mode with emission diameter bounds
at 0.01-0.1 μm, 0.1-1.0 μm, and 1.0-10.0 μm, respectively. Dust emission is parameterized following Zender
et al. (2003a). A geomorphic source function is used to account for global variations in soil erodibility, which
is proportional to the upstream runoff collection area (Zender et al., 2003b). The size distribution of emitted
dust particles follows the brittle fragmentation theory (Kok, 2011) with prescribed mass fractions of
0.00165%, 1.1%, and 98.9% for the three modes, respectively.

For comparison, we also use CESM1.2 (Hurrell et al., 2013) with CAM5 (Neale et al., 2010) and CLM4

(Oleson et al., 2010) as the atmosphere and land component, respectively. As shown in Table 1, the
representation of dust in aerosol module, dust emission scheme, and size distribution in CESM2.1 is the same
as in CESM1.2. The main difference of dust treatment is that CESM2.1 reduces the geometric standard
deviations in the accumulation and coarse mode, from 1.8 to 1.6 and 1.2, respectively. This greatly reduces
the dry deposition velocities for dust particles in the accumulation and coarse mode, which further leads to
the decrease of dust dry deposition fluxes. The geomorphic source function used in CESM2.1 is also different
from the one used in CESM1.2 (see Fig. S1), which substantially changes the spatial distributions of dust
emission.

**2.1.2 E3SM**

We use E3SMv1 (Golaz et al., 2019) with the atmosphere model (EAM, Rasch et al., 2019) and land

model (ELM), which are based on CAM5 and CLM4.5, respectively, as the atmosphere and land component.
Compared with CAM6, EAMv1 includes new treatments of convective transport, wet removal, and
resuspension of aerosols to the coarse mode (Wang et al., 2013, 2020), which can reduce the high model





biases of dust extinction in the upper troposphere. Dust is carried in the accumulation and coarse mode with
emission diameter bounds at 0.1-1.0 μm, and 1.0-10.0 μm, respectively. Unlike CESM1.2 and CESM2.1, the
size distribution of emitted dust particles follows Zender et al. (2003a) with prescribed mass fractions of 3.2%
and 96.8% for the accumulation and coarse mode, respectively (see Table 1). The higher mass fraction of
emitted accumulation mode dust in E3SMv1, which is three times larger than that in CESM2.1, can increase
the dust transport to remote regions (e.g., Arctic, Antarctic, and Southern Ocean). However, it overestimates
the mass fraction of emitted find dust compared with observations, as shown in Kok (2011). E3SMv1 uses
the same source function as CESM1.2 for dust emission, indicating that E3SMv1 has similar spatial
distributions of dust emission to CESM1.2. Compared with CESM1.2 and CESM2.1, E3SMv1 has 72 vertical
layers and its bottom layer thinner than that in CESM1.2 and CESM2.1, which can affect the dry deposition
of dust.

**2.1.3 MERRA-2**
MERRA-2 (Gelaro et al., 2017) is the latest atmospheric reanalysis of the modern satellite era produced
by combining GEOS atmospheric model version 5 (GEOS-5) with a 3D variational data assimilation
(3DVAR) algorithm to ingest a wide range of observational data. MERRA-2 assimilates AOD from the
Advanced Very High Resolution Radiometer (AVHRR), MODIS, MISR, and AERONET. GEOS-5 is run
with GOCART aerosol module (Chin et al., 2002). The dust emission flux is calculated based on Ginoux et
al. (2001). A topographic source function (see Fig. S1) is used to shift dust emission towards the most erodible
regions, which is characterized by the relative elevation of source regions in surrounding basins (Ginoux et
al., 2001). We should note that the assimilation of AOD results in the imbalance of global dust mass. Because
the assimilation of AOD increases dust concentrations in remote regions, the total deposition (dry and wet)



is considerably larger than the dust emission in MERRA-2. As shown in Table 1, dust is carried in 5 size bins
with diameter bounds at 0.2-2.0 μm, 2.0-3.6 μm, 3.6-6.0 μm, 6.0-12.0 μm, and 12.0-20.0 μm, respectively.
The size distribution of emitted dust particles follows Tegen and Lacis (1996) with mass fractions of 6.6%,
20.6%, 22.8%, 24.5%, and 25.4%, respectively. MERRA-2 includes very coarse dust (10.0-20.0 μm), which
is neglected by CESM and E3SM. MERRA-2 also has the highest mass fraction of emitted fine dust (0.1-10
μm) among the four models (see Figure 3 in Kok 2011), which can increase the dust transport.

**2.2 Experiments Design**
We ran CESM1.2 and CESM2.1 with the finite-volume (FV) dynamical core for CAM5.3 and CAM6,
respectively, at 0.9°×1.25° horizontal resolution with 56 vertical levels from 2006 to 2009, and the last 3-
year results were used for analysis. We ran E3SMv1 with the spectral-element (SE) dynamical core for
EAMv1 at 100 km horizontal resolution on a cubed-sphere geometry with 72 vertical layers from 2006 to
2009. The horizontal wind components u and v were nudged towards the MERRA-2 meteorology using a
relaxation time scale of 6 hours. Monthly mean climatological SST and sea ice concentrations were used.
The dust emission in CESM1.2, CESM2.1, and E3SMv1 was tuned so that AOD in the dusty region
(DOD/AOD>0.5) matches observations from MODIS onboard Terra and Aqua.

**2.3 Satellite Retrievals**
**2.3.1 MODIS and MISR**
Pu and Ginoux (2016) derived DOD over land from MODIS Collection 6 (C6) Deep Blue aerosol
products (Hsu et al., 2013) by using a continuous function relating the Ångström exponent ($\alpha$) to fine mode
AOD established by Anderson et al. (2005) which was derived based on ground measurements. The formula





is given as:
$$DOD = AOD \times (0.98 - 0.5089\alpha + 0.0512\alpha^2) \quad (\alpha < 0.3, \omega < 1) \tag{1}$$
where $\omega$ is the single scattering albedo at 470 nm. DOD is derived only when α is less
than 0.3 and ω is less. As discussed in Baddock et al. (2016), we use the lowest quality (QA=1) AOD over dust source
regions and AOD flagged as very good quality (QA=3) for other land areas. Although the derived MODIS
DOD over land is in good agreement with coarse mode AOD from AERONET (Pu and Ginoux, 2016), it
may overestimate DOD in reality. We calculate coarse mode AOD, which is used as a proxy of DOD, only
when AOD is mainly contributed by dust (α<0.3, ω<1).
Yu et al. (2019) derived DOD over ocean from MODIS C6 Dark Target aerosol products as follows:
$$DOD = \frac{AOD(f_c-f) - AOD_m(f_c-f_m)}{(f_c-f_d)} \tag{2}$$
where $f$ is the fine mode fraction retrieved directly from MODIS; $AOD_m$ is the marine AOD; $f_c$, $f_d$, and $f_m$ are
fine mode fractions of combustion, dust, and marine aerosol, respectively. $F_c$, $f_d$, and $f_m$ are set to be 0.92
(0.89), 0.26 (0.31), and 0.55 (0.48) for MODIS onboard Terra (Aqua), respectively. These differences in the
fractions may be caused by the difference in instrument calibrations (Levy et al., 2018). We also use the
nonspherical fraction of AOD from MISR (Witek et al., 2018) as a proxy of DOD over ocean (e.g., Kim et
al., 2014, 2019; Yu et al., 2019). We do not use MODIS and MISR DOD over high-latitude regions (> 60°)
because of large uncertainties in retrievals.

**2.3.2 CALIOP**
Luo et al. (2015a) developed a new dust separation method which derives the dust backscatter coefficient
($\beta_d$, $m^{-1}$ $sr^{-1}$) in the lidar equation inversion stage using the CAL-L1B data. The original single-scattering
lidar equation is:



$$\beta'(z) = \big(\beta_a(z) + \beta_m(z)\big)e^{-2\int_0^z \big(S_a\beta_a(z') + S_m\beta_m(z')\big)dz'}$$ (3)
where $\beta'$ (CAL-L1B product) is the total attenuated backscatter coefficient; $\beta_a$ (CAL-L2 product) and $\beta_m$ are
backscatter coefficients for aerosol and molecules, respectively; $S_a$ and $S_m$ are lidar ratios for aerosol and
molecules, respectively. Assuming that dust is externally mixed with non-dust aerosols, Eq. (3) can be
rewritten as:
$$\beta'(z) = \big(\beta_d(z) + \beta_{nd}(z) + \beta_m(z)\big)e^{-2\int_0^z \big(S_d\beta_d(z') + S_{nd}\beta_{nd}(z') + S_m\beta_m(z')\big)dz'}$$ (4)
where $\beta_d$ and $\beta_{nd}$ are backscatter coefficients for dust and non-dust aerosols, respectively; $S_d$ is the lidar ratio
for dust and set to be 40 sr; $S_{nd}$ is the lidar ratio for non-dust aerosols and set to be 25 sr. The new separation
method also requires a priori knowledge of depolarization ratios of dust ($\delta_d$) and non-dust ($\delta_{nd}$), which are
given values of 0.25 and 0.05, respectively. The dust extinction can then be easily converted from $\beta_d$ by
multiplying $S_d$ of 55 sr, which accounts for the multiple scattering effects as suggested in Wandinger et al.
(2010). The new separation method can resolve dust extinction from polluted dust (i.e. dust mixing with other
types of aerosols), whereas CAL-L2 products fail to do so. It also tends to have less uncertainties than doing
the partition based on lidar inversion products (i.e., CAL-L2) in previous studies (e.g., Amiridis et al., 2013;
Proestakis et al., 2018; Yu et al., 2015). Additionally, Luo et al. (2015b) developed a new dust identification
method by using combined lidar-radar cloud masks from CloudSat and CALIPSO, which significantly
improves the detection of optically thin dust layer, especially in the upper troposphere. In this study, we use
both the new separation method (Luo et al., 2015a) and the new identification method (Luo et al., 2015b) to
produce the nighttime dust extinction dataset (L15) for the period of 2007 to 2009.
Yu et al. (2015) derived $\beta_d$ from CAL-L2 $\beta_a$ with a priori knowledge of $\delta_d$ and $\delta_{nd}$ as follows:
$$\beta_d = \frac{(\delta - \delta_{nd})(1 + \delta_d)}{(1 + \delta)(\delta_d - \delta_{nd})}\beta_a$$ (5)
where $\delta$ is the CALIOP observed particulate depolarization ratio. To minimize the uncertainties, we calculate


$\beta_d$ in two scenarios: the "lower-bound dust fraction" scenario ($\delta_d$=0.30, $\delta_{nd}$=0.07) and the "upper-bound dust
fraction" scenario ($\delta_d$=0.20, $\delta_{nd}$=0.02). We then converted dust extinction from $\beta_d$ by multiplying $S_d$ of 45 sr.
In this study, we use the dust separation method to retrieve nighttime dust extinction under the cloud free
condition based on CAL-L2 version 4 lidar products. To ensure the retrieval quality, we only select high-
confidence data based on the cloud-aerosol discrimination (CAD) scores (-100 to -70) and extinction quality
control flag values (0, 1, 16, and 18) (Yu et al., 2010; Yu et al., 2015). The aerosol free condition (dust
extinction is zero) is also included in the retrieval.
To make an apple-to-apple comparison of modeled dust extinction with satellite observations, two
treatments were applied to collocate model results and CALIOP data. First, dust extinction retrievals from
L15 and Y15 were averaged into 0.9º×1.25º grid boxes (same as CAM5.3 and CAM6) and interpolated to
pressure levels at 25 hPa intervals. Modeled dust extinction profiles from CESM1.2, CESM2.1, and E3SMv1
were sampled every 10 s along the CALIPSO satellite tracks. Dust extinction profiles from MERRA-2 were
calculated offline based on 3-hourly output of 3-D dust mixing ratio and then sampled along the CALIPSO
satellite tracks. Second, the dust extinction in and below the vertical layer where cloud fraction is 100% was
set to missing values to account for the fact that dust inside clouds, adjacent to the cloud bottom, and bellow
optically thick clouds cannot be retrieved from CALIOP. Collocated dust extinction from model experiments
is then integrated vertically to get the DOD value.

**3 Results**
Figure 1a shows 12 selected regions including both dust source regions and transport pathway regions,
in which we evaluate the seasonal variations of modeled dust extinction and DOD with satellite retrievals.
Figure 1b shows the network of stations, at which we evaluate dust surface concentrations (Huneeus et al.,



2011; Prospero et al., 2012; Fan, 2013).

**3.1 Dust Mass Budgets**

Table 2 gives the global annual mean dust mass budgets, DOD, and mass extinction efficiency (MEE)

from model experiments. We can see that dust emissions in CESM1 and E3SMv1 are much larger than those
in CESM2 and MERRA-2, which can be attributed to the model tuning and uses of different dust emission
schemes and source functions. Dust emission schemes in CESM1, CESM2, and E3SMv1 are the same and
based on Zender et al. (2003a), while dust emission scheme in MERRA-2 is based on Ginoux et al. (2001).
CESM1 and E3SMv1 use the same dust source function which is different from those in CESM2 and
MERRA-2. Dry deposition is the dominant removal process of dust compared with wet deposition in CESM1,
E3SMv1, and MERRA-2, whereas CESM2 has less dry deposition (675 Tg yr$^{-1}$) than wet deposition (1151
Tg yr$^{-1}$). Due to the decrease of geometric standard deviations in the accumulation and coarse mode of
CESM2 MAM4 (see Table 1), aerosol dry deposition velocities for the accumulation and coarse mode greatly
reduce, leading to the decrease of dry deposition. Note that MERRA-2 has less dry deposition (750 Tg yr$^{-1}$)
than wet deposition (865 Tg yr$^{-1}$) for dust aerosols with diameter between 0.2 and 12.0 μm. We also find that
E3SMv1 produces notably higher dry deposition than CESM1, although both models have similar amount of
dust emission. In CESM and E3SM, dust emission fluxes (kg m$^{-2}$ s$^{-1}$) are divided by the model bottom layer
thickness and converted to dust mixing ratio tendencies (kg kg$^{-1}$ s$^{-1}$). Because the bottom layer in E3SMv1
is thinner with higher vertical resolution than the one in CESM1, more dust in the bottom layer is removed
through dry deposition process.

As CESM2 has much less dust dry deposition than wet deposition, larger fraction of dust is transported

away from the major source regions in CESM2 than CESM1. Dust lifetime in CESM2 (3.90 days) is longer



than that in CESM1 (2.33 days). E3SMv1 has a smaller dust burden and a shorter lifetime but larger DOD
than CESM1 due to the larger dry deposition and higher mass fraction of dust in the accumulation mode,
respectively. Since MERRA-2 has the largest mass fraction of fine dust among the four models and
assimilates AOD, dust in MERRA-2 has the longest lifetime (4.19 days) and largest global mean DOD
(0.0312), despite its lowest dust emission. Note that MERRA-2 has considerably larger dust deposition (dry
and wet, 2048 Tg yr$^{-1}$) than dust emission (1636 Tg yr$^{-1}$), which is significantly imbalanced, due to the
assimilation of AOD. In remote regions where AOD is underestimated, the assimilation of AOD increases
dust concentrations resulting in the increase of dust deposition. MEE (DOD/dust burden) is often used for
converting dust mass to DOD. As shown in Table 2, it varies from 0.452 (CESM1) to 0.677 m$^2$ g$^{-1}$ (MERRA-
2). In Huneeus et al. (2011), MEE from AeroCom Phase I models varies from 0.25 to 1.28 m$^2$ g$^{-1}$. Haywood
et al. (2003) measured MEE of 0.37 m$^2$ g$^{-1}$ (0.32-0.43 m$^2$ g$^{-1}$) based on aircraft campaigns, which is used in
many studies (e.g., Kaufman et al., 2005; Yu et al., 2015). Pu and Ginoux (2018) used a MEE of 0.6 m$^2$ g$^{-1}$
to convert dust burden simulated by CMIP5 models to DOD.
Figure 2 shows the spatial distributions of global annual mean dust emissions from the model experiments.
We can see that CESM1 (Fig. 2a) has similar spatial distributions of dust emission as E3SMv1 (Fig. 2c) due
to the use of the same source function and dust emission scheme. Dust emission in MERRA-2 (Fig. 2d)
spreads more uniformly than that in CESM1 and E3SMv1, while CESM2 (Fig. 2b) has smaller areas emitting
mineral dust than CESM1 and E3SMv1. CESM2 has lower dust emission in main source regions, such as
North Africa, Middle East, and East Asia, but has much higher dust emission in South America, South Africa,
and Australia than CESM1, E3SMv1, and E3SMv1.
Figure 3 shows the seasonal variations of dust emissions from model experiments in six source regions
(Fig. 1a). In North Africa (Fig. 3a), CESM1 has the largest dust emission (5000-10000 kt d$^{-1}$) with the



strongest seasonality, while CESM2 has the lowest dust emission (~2000 kt d$^{-1}$). Dust emissions in CESM1,
CESM2, E3SMv1, and MERRA-2 peak in April, February, February, and July, respectively. Although
CESM1 and E3SMv1 use the same source function and dust emissions scheme, E3SMv1 produces
considerably lower dust emission than CESM1. The different height of bottom layer with horizontal wind
nudged toward MERRA-2 may cause the differences in friction velocities. Large differences of dust emission
can also be found in Northwest China (Fig. 3b). However, dust emissions in the four models have similar
seasonality and all peak in May. E3SMv1 produces slightly higher dust emission than CESM1, especially
from September to January. CESM1, CESM2, and MERRA-2 produces similar low dust emissions in
December and January. In North America (Fig. 3d), CESM2 produces the lowest dust emission with the
weakest seasonality among the four models. In the Southern Hemisphere (SH) source regions (Fig. 3c, e and
f), CESM2 produces much larger dust emissions than CESM1, E3SMv1, and MERRA-2. In South America,
the seasonality of dust emission in CESM2 is significantly different from those in other models, which results
from the different location of dust emission (see Fig. 2).
Figure 4 shows the seasonal variations of dust burdens from model experiments in the twelve selected
regions marked in Fig. 1a. In North Africa (Fig. 4a), CESM1 has the highest dust burden while CESM2 has
the lowest dust burden. Although MERRA-2 produces much lower dust emission than E3SMv1, dust burden
in MERRA-2 is larger than that in E3SMv1 due to a higher mass fraction of fine dust. Because the
assimilation of AOD increases the dust concentrations on the trans-Atlantic pathway, MERRA-2 has the
highest dust burden among the four models across the Atlantic (Fig. 4e). In North America (Fig. 4i), dust
burden in MERRA-2 is much larger than those in other models, whereas dust emission in MERRA-2 is
similar to those in CESM1 and E3SMv1. This is due to the enhanced dust transport over the Pacific, which
is further caused by the assimilation of AOD over the Pacific (see Fig. 4f and j). We can see that CESM2





produces the highest dust burden with the strongest seasonality in SH source regions (Fig. 4c, g, and k) due
to its large dust emission. MERRA-2 has similar dust burden in the Arctic (Fig. 4d) as in Northwest China,
indicating that MERRA-2 may overestimate dust burden in the Arctic.

**3.2 Dust Optical Depth**
Figure 5 compares the spatial distributions of modeled DOD with satellite retrievals from CALIOP (82°S-
82°N), MODIS (60°S-60°N) and MISR (ocean, 60°S-60°N). The annual mean values are averaged between
60°S and 60°N for a better comparison. In general, CESM1, CESM2, and E3SMv1 underestimate global
mean DOD compared with CALIOP (L15 and Y15) and MODIS; DOD in MERRA-2 is higher than CALIOP
but is still much lower than MODIS DOD. CESM1 overestimate the land DOD (0.0678) compared with
observations from L15 (0.0614) and Y15(0.0625); DOD over land in E3SMv1 (0.0615) is between L15 and
Y15. However, modeled DOD over ocean in CESM1 (0.0074), CESM2 (0.0087), and E3SMv1 (0.0094) is
much lower than retrievals from L15 (0.0137) and Y15 (0.0181), which mainly contributes to the low model
biases of global mean DOD. This indicates that CESM1, CESM2, and E3SMv1 underestimate dust transport
to remote regions (e.g., Arctic and Southern Ocean). In the Northern Hemisphere (NH), CESM2 produces
the lowest DOD over major source regions such as North Africa, Middle East, and East Asia among the four
models due to its low dust emission. Since E3SMv1 has higher mass fraction (3.2%) of accumulation mode
dust than CESM1 and CESM2 (1.1%), it performs better than CESM1 and CESM2 and simulates more dust
transport to the Arctic. In SH, CESM2 produces much larger DOD in South America, South Africa, and
Australia than CALIOP due to high dust emission in these three source regions (see Fig. 3), which also leads
to a higher DOD over the Southern Ocean than other models and improves the agreement with observations.
MEERA-2 tends to have the best agreement with CALIOP in DOD, especially in remote regions, which can





be attributed to the assimilation of AOD from satellites and ground-based measurements and high mass
fraction of emitted fine dust.

Comparing to the DOD estimates from AeroCom models ($0.028 \pm 0.011$, Huneeus et al., 2011) and Ridley

et al. (2016) ($0.030 \pm 0.005$), global mean DOD in MERRA-2 and Y15 is close to the global mean value from
Ridley et al. (2016); DOD from model experiments is within the uncertainty range of AeroCom models.
MODIS DOD ($> 0.06$) is substantially larger than CALIOP DOD ($< 0.03$). MISR DOD over ocean is between
CALIOP and MODIS DOD. Large uncertainties also exist in DOD retrievals from different sensors, which
can affect the model evaluation. The DOD differences between MODIS and CALIOP can come from two
main aspects: (1) the differences between AOD retrieved from MODIS and CALIOP and (2) the differences
of retrieval algorithms in separating DOD from AOD. Ma et al. (2013) compared CAL-L3 AOD with MODIS
AOD from 2006 to 2011 and found that CAL-L3 AOD is lower than MODIS AOD. Global annual mean
AOD from nighttime CAL-L3 over ocean is 0.089, while MODIS AOD over ocean is 0.148 and 0.140 for
Terra and Aqua, respectively. Ma et al. (2013) also showed that CAL-L3 has lower AOD than MODIS over
major dust source regions.

MODIS DOD is subject to cloud contamination that can cause a high bias in DOD (e.g., Zhang et al.,

2005). In Fig. 5g and h, we can see the apparent discontinuity along the tropical African coast, because
MODIS DOD is derived from Deep Blue and Dark Target products over land and ocean, respectively. In
addition, MODIS DOD derived from Dark Target products over the turbid-water coastal region is subject to
high bias due to the underestimation of surface reflectance. Since Eq. (1) is used to calculate the coarse mode
AOD in Anderson et al. (2005) and we derived DOD only when AOD is mainly contributed by dust ($\alpha < 0.3$,
$\omega < 1$), MODIS DOD over land may be subject to high bias. Unlike passive sensors, CALIOP may do a better
job in discriminating dust from clouds and other types of aerosols and providing the vertical distributions of



dust. However, CALIOP cannot penetrate optically thick cloud layers due to strong attenuation of the signals,
missing the lowest part of aerosol plumes. CALIOP also fails to detect tenuous dust layers due to weak signals.
Notable differences are found between MODIS DOD from Terra (0.0686) and Aqua (0.0615) as well, which
can be attributed to the calibration issues of MODIS Terra (e.g., Levy et al., 2018). Ma and Yu (2015) showed
that MISR AOD over ocean (0.157) is higher than MODIS Aqua AOD over ocean (0.139). MISR DOD over
ocean, especially over the Southern Ocean, may be biased high due to artifacts (e.g., Witek et al., 2013). In
this study, we use the latest version (V23) of MISR aerosol products, which significantly reduces AOD over
ocean compared to the previous V22 products.
Table 3 gives the global seasonal mean DOD (averaged over 60°S-60°N) from model experiments and
satellite observations. CESM1, CESM2, and E3SMv1 underestimate global mean DOD in all seasons
compared with MODIS and CALIOP, which is mainly attributed to the low model biases of DOD over ocean.
DOD from model experiments, Y15, and Terra MODIS all peaks in MAM (March-April-May) and reaches
its minimum in DJF (December-January-February) due to the seasonal variations of global dust emission.
However, DOD from L15 and Aqua MODIS slightly increases from MAM to JJA (June-July-August) and
peaks in JJA. Notable decreases of DOD from MAM to JJA are found in model experiments. The decrease
ranges from 0.0012 (E3SMv1) to 0.0096 (MERRA-2), while DOD from Terra MODIS and Y15 slightly
decreases by 0.0008 and 00019, respectively. Unlike observations and other models, DOD from CESM2
increases from JJA to SON (September-October-November) which can be attributed to the overestimation of
dust emission in SH. CESM2 also has the weakest seasonal contrast, and the DOD difference between MAM
and DJF is only 0.0067. MERRA-2 has the strongest seasonal contrast, and the DOD difference between
MAM and DJF is 0.0244.
We further examine the dust transport across the Atlantic (0°-35°N) and Pacific (30°N-60°N) by



comparing the meridional means of modeled DOD with satellite retrievals from CALIOP, MODIS (combined
Terra and Aqua), and MISR, as shown in Fig. 6. In Fig. 6a, satellite retrievals of DOD show high values in
North Africa (15°W-30°E). As dust is transported from North Africa to the Atlantic, DOD gradually decreases.
In the source regions, MODIS and CALIOP DOD all peaks between 5°W and 5°E, whereas DOD from
CESM1, CESM2, and E3SMv1 peaks in Northeast Africa (30°E) determined by the geomorphic source
function used in the models. Although MERRA-2 well captures the meridional variations of DOD due to the
use of a topographic source function, it overestimates the DOD compared with CALIOP. This may be caused
by the contribution of very coarse dust (10-20 μm) and high mass fraction of fine dust (0.1-1μm). DOD in
E3SMv1 agrees the best with CALIOP DOD among the four models. CESM1 produces substantially larger
DOD (0.25-0.38) in Northeast Africa (15°E -30°E) than CALIOP but agrees well with CALIOP in Northwest
Africa (15°W-5°E). CESM2 significantly underestimates DOD (~0.1) in Northwest Africa (15°W-5°E)
compared with CALIOP due to its underestimation of dust emission (see Fig. 3a).

Over the entire Atlantic, modeled DOD in CESM1, CESM2, and E3SMv1 is lower than observations,

which may result from the fast deposition and short lifetime (see Table 2). E3SMv1 performs better than
CESM1 and CESM2 because of its higher mass fraction of fine dust. Although DOD in MERRA-2 agrees
well with CALIOP DOD over the Atlantic, it tends to have much faster drop than CALIOP along the transport
pathway, especially between 20°W and 0°. This suggests that dust in MERRA-2 may also deposit too fast.
The decline rate of DOD in E3SMv1 agrees well with that in CALIOP. Because of the reduced geometric
standard deviation in the coarse mode in CESM2 (Table 1), dust dry deposition decreases, and dust lifetime
increases significantly, which explains the weak longitudinal gradient of DOD in CESM2. Similar
conclusions can be drawn from Fig. 6b for dust transport across the Pacific. CESM1, CESM2, and E3SMv1
underestimate DOD over the Pacific but overestimate DOD in source regions (i.e., Taklamakan and Gobi



Desert) of East Asia compared with CALIOP. DOD from MERRA-2 is higher than CALIOP over both East
Asia and the Pacific. Large disparities of DOD from CALIOP, MODIS, and MISR are found over both land
and ocean. CALIOP DOD is lower than MODIS DOD, and the differences are larger over land (~0.1). MISR
DOD over ocean is close to CALIOP DOD over the Atlantic and MODIS DOD over the Pacific.
Figure 7 shows the seasonal variations of modeled DOD in comparison with satellite retrievals from
CALIOP, MODIS, and MISR at 12 selected regions. In North Africa (Fig. 7a), CESM2 significantly
underestimates DOD in MAM, JJA, and SON due to its low dust emission (see Figs. 3a and 4a). DOD in
E3SMv1 agrees well with CALIOP DOD, while CESM1 and MERRA-2 overestimates DOD in all seasons
compared with CALIOP. Over the Atlantic (Fig. 7e), DOD in MERRA-2 agrees well with CALIOP DOD in
all seasons, while E3SMv1 underestimates DOD in MAM and JJA. This suggests that wet removal of dust
in E3SMv1 over the Atlantic in MAM and JJA may be too strong. In North America (Fig. 7i), CESM1,
CESM2, and E3SMv1 produces much lower DOD due to the underestimation of dust transport across the
Pacific. MODIS DOD peaks in July similar to the seasonality of trans-Atlantic dust transport, while CALIOP
DOD peaks in May similar to the seasonality of trans-Pacific dust transport. Unlike North Africa, all models
overestimate DOD in MAM, JJA, and SON compared with CALIOP in Northwest China (Fig. 7b) due to
overestimation of dust emission. Because E3SMv1 has larger dust emission than CESM1 and CESM2 in DJF
(Fig. 3b), the low bias of DOD is improved. This suggests that CESM1 and CESM2 may underestimate dust
emission in DJF over Northwest China. Over the Pacific (Fig 7f and j), DOD in E3SMv1 agrees well with
CALIOP DOD from May to October, while CESM1 and CESM2 underestimate DOD in all seasons,
especially in DJF by over one order of magnitude. DOD in all models and MODIS reaches its minimum in
December or January, whereas CALIOP DOD has its minimum in August.
Figure 7c, g, and k focus on the source regions in SH. The seasonal variations of DOD in SH are opposite



to NH due to opposite seasons in SH. CESM2 significantly overestimates DOD in all seasons compared with
CALIOP, by one order of magnitude due to the overestimation of dust emission, while CESM1, E3SMv1,
and MERRA-2 perform reasonably well. Figure 7d, h, and l focus on the three remote regions where the
largest disagreements between model simulations and observations are found. In the Arctic (Fig. 7d), CESM1,
CESM2, and E3SMv1 all have low biases of DOD, but E3SMv1 performs better than CESM1 and CESM2,
especially in DJF. CESM2 performs slightly better than CESM1 due to the reduced geometric standard
deviations in the accumulation and coarse mode. MERRA-2 overestimates DOD compared with CALIOP
due to excessive dust transport from NH source regions. Over the tropical Pacific (Fig. 7h), CALIOP, MODIS,
and MISR DOD all shows small seasonal contrast, while MERRA-2 shows considerable seasonal contrast of
DOD with its maximum in May and its minimum in November, which is influenced by dust transport over
the North Pacific. In the Southern Ocean (Fig. 7l), MODIS and MISR DOD has much stronger seasonal
variations than CALIOP DOD. Because of the assimilation of AOD, MERRA-2 also has opposite seasonal
variations to CALIOP DOD as MODIS and MISR. The difference in the seasonality of retrieved DOD may
come from cloud contamination over the Southern Ocean. In the selected regions, DOD from Y15 is generally
larger than that from L15, because the differences in retrieval algorithms lead to higher dust extinction in the
lower troposphere for Y15.

**3.3 Dust Extinction**

Figure 8 compares annual mean vertical profiles of modeled dust extinction with satellite retrievals from

L15 and Y15 in 12 selected regions. In North Africa (Fig. 8a), modeled dust extinction agrees well with
observations from L15 and Y15 in the lower and middle troposphere (> 500 hPa). In the upper troposphere
(< 400 hPa), significant high model biases of dust extinction are found in all models and over one order of





magnitude in CESM1 and MERRA-2, which comes from JJA and SON (see Figs. S2-S5). It is likely due to
excessive convective transport (e.g., Allen & Landuyt, 2014) and lack of secondary activation of aerosols
entrained into convective updrafts (e.g., Wang et al., 2013; Yu et al., 2019) in the models. As E3SMv1 uses a
unified aerosol convective transport scheme with secondary activation (Wang et al., 2013, 2020), the high
model biases of dust extinction are reduced. Due to its lower dust emission in North Africa (Fig. 3a), less
dust is lifted up throughout the troposphere in CESM2 than in the other models. MERRA-2 has the largest
high biases of dust extinction in the upper troposphere because of its highest fine mode mass fraction. As
dust is transported to the Atlantic, the dust extinction decreases at all levels (Fig. 8e). Dust extinction in
E3SMv1 agrees well with CALIOP. CESM1 underestimates dust extinction below 500 hPa but overestimates
dust extinction above 500 hPa. MERRA-2 agrees well with the observations below 500 hPa but is much
larger than observations in the upper troposphere. In North America (Fig. 8i), CESM1, CESM2, and E3SMv1
greatly underestimate dust extinction in the lower troposphere by one order of magnitude. The low model
biases reach the maximum in JJA (Fig. S3) and the minimum in DJF (Fig. S5). Since MERRA-2 has similar
dust emission as CESM1 and E3SMv1 but only slightly underestimates dust extinction in the lower
troposphere. The low biases of dust extinction in CESM1, CESM2, and E3SMv1 are mainly caused by the
underestimation of dust transport across the Pacific. We can see that in the Northeast Pacific (Fig. 8j),
MERRA-2 and L15 still has dust extinction of 0.001-0.002 km$^{-1}$ in the bottom layer. The high biases of dust
extinction in MERRA-2 above 600 hPa are consistent with the overly strong transport across the Atlantic and
Pacific.

As shown in Fig. 8b, f, and j, CESM1, CESM2, and E3SMv1 have high biases of dust extinction in

Northwest China but low biases over the Pacific. The magnitude of the low biases of dust extinction peaks
in DJF (Fig. S5), which corresponds to the low biases of DOD in Fig. 7. CALIOP dust extinction profiles



vary little across the Pacific, while dust extinction at all levels in CESM1, CESM2, and E3SMv1 decreases
notably, resulting in the increase of low biases of DOD with distance from the source. MERRA-2
overestimates dust extinction above 800 hPa over the Pacific and shows a slightly increase from 1000 hPa to
600 hPa. This indicates that MERRA-2 significantly overestimates the dust transport across the Pacific.
CESM2 significantly overestimates dust extinction at all levels in the three SH source regions (Fig. 8c, g,
and k) due to the overestimation of dust emission. In South America, CESM1 and E3SMv1 underestimate
dust extinction below 900 hPa. This suggests that the two models may underestimate the dust emission. In
the Arctic (Fig. 8d), E3SMv1 improves dust extinction at all levels compared with CESM1, while CESM2
only increases dust extinction below 800 hPa. Over the Southern Ocean, CESM1, CESM2, and E3SMv1 all
underestimate dust extinction below 850 hPa and produce an increase compared to the bottom level. The
overestimation of dust extinction above 800 hPa by MERRA-2 is also evident in Fig. 8d, h, and l. We note
that there are considerable differences between satellite retrievals from L15 and Y15. Dust extinction from
L15 is larger in the upper troposphere and lower in the lower troposphere than that from Y15, which is due
to different dust identification and separation methods (Wu et al., 2019).

**3.4 Dust Surface Concentration**
Figure 9 compares simulated annual mean dust surface concentrations with observations at 24 sites, as
shown in Fig. 1b. We use the dust surface concentrations for 0.2-12 μm (bins 1-4) in MERRA-2 for better
comparison with CESM1, CESM2, and E3SMv1. Note that all measurements of dust surface concentrations
except for observations at Barbados and Miami were conducted prior to 2007-2009. Some observations are
derived from measurements of aluminum by assuming a certain fraction. CESM1, CESM2, and E3SMv1
have low biases, while MERRA-2 has high biases at most sites. E3SMv1 performs better than CESM1 and



CESM2 in terms of the overall correlation (R), mean bias (MB), and mean normalized bias (MNB). CESM2
has the lowest correlation and the highest overall MB and MNB. The overall underestimation of dust surface
concentrations in CESM1, CESM2, and E3SMv1 mainly results from the low biases at sites in the Arctic,
Antarctic, and Tropical Pacific.

Figure 10 shows the seasonal variations of modeled dust surface concentrations in comparison with

observations at 12 selected sites. At Izana (Fig. 10a) which is close to the west coast of North Africa, all
models underestimate dust surface concentrations due to low dust emissions in Northwest Africa (15°W-5°E)
and fail to capture the seasonality. Although DOD in MERRA-2 agrees well with CALIOP observations over
the Atlantic (see Fig. 6a), MERRA-2 still has considerable low biases in dust surface concentrations because
of too much dust emitted in the fine mode. Dust surface concentrations in the four models agree better with
observations at Barbados (Fig. 10e) than at Miami (Fig. 10i). CESM1, CESM2, and E3SMv1 underestimate
dust surface concentrations at Miami, especially in DJF by more than one order of magnitude. E3SMv1 tends
to have the best agreement with observations at Cheju (Fig. 10b), while CESM1 and CESM2 have strong
low biases in JJA and DJF. MERRA-2 overestimates the concentrations at Midway Island and Oahu Hawaii
in all months.

Figure 10c, g, and k show three sites in NH high-latitude regions. E3SMv1 significantly improves the

dust surface concentrations compared with CESM1 and CESM2 at Alert, but it still has low biases, especially
in SON and DJF by one order of magnitude. Ground measurements show high dust surface concentrations
in SON due to local dust emission in NH high-latitude regions (Fan et al., 2013; Groot Zwaaftink et al., 2016),
but CESM1, CESM2, and E3SMv1 miss the local dust sources there. CESM1 and E3SMv1 tend to have
stronger low model biases of dust surface concentrations at Heimaey than at Alert, while CESM2 tend to
have weaker low model biases at Heimaey than at Alert, especially in DJF. Figure 10d, h, and l show three





sites in the Tropical Pacific and Antarctic. At Palmer Station, CESM1 underestimates dust surface
concentrations by three orders of magnitude. Dust surface concentrations in CESM2 are higher than CESM1
and E3SMv1 due to higher dust emission in SH and the reduced geometric standard deviations. Because
E3SMv1 produces small amount of dust emission in the Antarctic (Fig. 2c), it also has higher concentrations.

**4 Discussion and Conclusions**
In this study, we evaluate the spatiotemporal variations of dust extinction profiles and DOD in CESM1,
CESM2, E3SMv1, and MERRA-2 against satellite retrievals from CALIOP (L15 and Y15), MODIS, and
MISR. We find that CESM1, CESM2, and E3SMv1 underestimate global annual mean DOD compared with
CALIOP and MODIS, which can be mainly attributed to the low model biases of DOD over ocean. This
indicates that CESM1, CESM2, and E3SMv1 underestimate dust transport to remote regions. E3SMv1
performs better than CESM1 and CESM2 in NH due to its higher fine mode mass fraction of dust. CESM2
performs the worst in NH due to its lower dust emission but improves DOD in SH due to its high dust
emissions in SH source regions. DOD in MERRA-2 agrees well with CALIOP DOD in remote regions due
to the assimilation of AOD and its higher mass fraction of fine mode dust. All models tend to overestimate
dust extinction in the upper troposphere of source regions because of excessive convective transport and/or
lack of secondary activation of aerosols entrained into convective updrafts. The latter is considered in
E3SMv1 (Wang et al., 2020), which thus shows a reduced bias of dust extinction in the upper troposphere.
The high model biases of dust extinction in MERRA-2 in the upper troposphere are persistent around the
globe.
CESM1, CESM2, and E3SMv1 produce substantial greater DOD than CALIOP in Northeast Africa and
fail to capture the spatial distributions of DOD in North Africa, which can be significantly improved by using





the source function of Ginoux et al. (2001) or the dust emission scheme of Kok et al. (2014a, 2014b) (K14).
The three models also overestimate DOD over Northwest China due to the overestimation of dust emission
in MAM, JJA, and SON. Wu et al. (2019) showed that CESM1 with K14 dust emission scheme better agrees
with CALIOP observations in Northwest China. Since the source functions used in the four models are all
zeros north to 60°N, the four models don't produce any dust emissions in NH high-latitude regions, while
ground observations indicate considerable local dust sources.
The low model biases of DOD over the Atlantic in CESM1, CESM2, and E3SMv1 can be greatly
improved if the high dust emission in Northeast Africa is captured by models. E3SMv1 has similar decline
rate of DOD as CALIOP from Northeast Africa to the Atlantic. CESM1, CESM2, and E3SMv1 underestimate
DOD in remote regions resulting from too fast dust deposition. Wu et al. (2018) showed that lower dry
deposition velocities for fine particles results in higher dust concentrations in remote regions (see Figure S1).
Dust emission in the three models is only added to the bottom layer currently, while dust storms in reality
can bring dust to high altitudes. The turbulent mixing of dust in the boundary layer needs to be improved.
Substantial differences are also found between MODIS and CALIOP DOD, which can greatly affect
model evaluation. MODIS DOD (> 0.06) is significantly larger than CALIOP DOD (< 0.03). DOD over
ocean from MISR is between MODIS and CALIOP. The differences between MODIS and CALIOP DOD
may come from instrument differences, artifacts such as cloud contamination and calibration issues, and
different retrieval algorithms. Ground lidar measurements, such as the Micro-Pulse Lidar Network
(MPLNET) and the European Aerosol Research Lidar Network (EARLINET), are needed to validate the
satellite retrievals from CALIOP, MODIS, and MISR.

**Code Availability**





The CESM1.2 source code is available at https://github.com/YamataSensei/CESM-code. The CESM2.1
source code is available at https://github.com/ESCOMP/cesm. The E3SMv1 source code is available at
https://github.com/E3SM-Project/E3SM.

**Data Availability**
The model output of CESM1 and CESM2 is archived at NCAR Cheyenne supercomputer. The model output
of E3SMv1 is archived at NERSC Cori supercomputer. MERRA-2 data is available at
https://disc.gsfc.nasa.gov/. CALIOP, MODIS and MISR data can be obtained online at
https://search.earthdata.nasa.gov.

**Author Contribution**
MW and XL conceived the project. MW designed and ran the model simulations with help and input from
XL, YS, CW, and ZK. HY, KY, TL, and ZW derived dust extinction profiles from CALIOP. AD derived dust
extinction profiles from MERRA-2. MW led the analysis and wrote the first draft of the paper. All coauthors
participated in discussions on data analysis and revised the paper.

**Competing Interests**
The authors declare that they have no conflict of interest.

**Acknowledgement**
This work is supported by NASA CloudSat and CALIPSO Science Program (grant NNX16AO94G). This
work is also funded by the U.S. Department of Energy (DOE), Office of Science, Office of Biological and





Environmental Research, Earth and Environmental System Modeling program as part of the Energy Exascale
Earth System Model (E3SM) project. The Pacific Northwest National Laboratory (PNNL) is operated for
DOE by Battelle Memorial Institute under contract DE-AC05-76RLO1830. We would like to acknowledge
the use of computational resources for conducting the model simulations at the National Energy Research
Scientific Computing Center (NERSC), a U.S. DOE Office of Science User Facility operated under contract
DE-AC02-05CH11231, and the NCAR-Wyoming Supercomputing Center provided by the NSF and the State
of Wyoming and supported by NCAR's Computational and Information Systems Laboratory. We would like
to thank Dr. Paul Ginoux for providing MODIS DOD over land, Dr. Joseph M. Prospero for providing the
measurements of dust surface concentrations at Heimaey, Barbados, and Miami, and Dr. Songmiao Fan for
providing the measurements of dust surface concentrations at Alert.

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





**Tables**
**Table 1.** Description of the models on their dust physical characteristics.

| | Resolution | Aerosol Module | Geometric Standard Deviations | Mass Fraction of Dust Emission (%) | Dust Emission Scheme |
|---|---|---|---|---|---|
| CESM1 | 1°, 30L | MAM4 (Liu et al., 2016) (3 modes, 0.01-0.1-1.0-10.0 μm) | 1.6, 1.8, 1.8 | 0.00165, 1.1, 98.9 (Kok, 2011) | Zender et al. (2003a) |
| CESM2 | 1°, 32L | MAM4 (Liu et al., 2016) (3 modes, 0.01-0.1-1.0-10.0 μm) | 1.6, 1.6, 1.2 | 0.00165, 1.1, 98.9 (Kok, 2011) | Zender et al. (2003a) |
| E3SMv1 | 1°, 72L | MAM4 (Liu et al., 2016) (2 modes, 0.1-1.0-10.0 μm) | 1.8, 1.8 | 3.2, 96.8 (Zender et al., 2003a) | Zender et al. (2003a) |
| MERRA-2 | 0.5°, 72L | GOCART (Chin et al., 2002) (5 bins, 0.2-2.0-3.6-6.0-12.0-20.0 μm) | | 6.6, 20.6, 22.8, 24.5, 25.4 (Ginoux et al., 2001) | Ginoux et al. (2001) |


















**Table 2.** Global annual mean dust mass budgets, DOD, and MEE

|  | CESM1 | CESM2 | E3SMv1 | MERRA-2 |
|---|---|---|---|---|
| Emission (Tg yr$^{-1}$) | 3868 (43, 3826) | 1820 (20, 1800) | 3399 (109, 3291) | 1636 (1220) |
| Dry deposition (Tg yr$^{-1}$) | 2496 (7, 2489) | 675 (5, 670) | 2638 (29, 2609) | 1168 (750) |
| Wet deposition (Tg yr$^{-1}$) | 1379 (36, 1343) | 1151 (15, 1136) | 764 (80, 684) | 880 (865) |
| Burden (Tg) | 24.7 (0.7, 24.0) | 19.5 (0.3, 19.2) | 17.9 (2.0, 15.9) | 23.5 (22.8) |
| Lifetime (day) | 2.33 (5.92, 2.29) | 3.90 (5.91, 3.88) | 1.92 (6.84, 1.76) | 4.19 (5.17) |
| DOD | 0.0219 | 0.0212 | 0.0238 | 0.0312 |
| MEE (m$^2$ g$^{-1}$) | 0.452 | 0.553 | 0.677 | 0.677 |

Note: the values in parentheses for CESM1, CESM2, and E3SMv1 correspond to the accumulation mode

(0.1-1 µm) and coarse mode (1-10 µm), respectively; the values in parentheses for MERRA-2 correspond to

bins 1-4 (0.2-12.0 µm)





**Table 3.** Global seasonal mean DOD (60°S-60°N)

|  | MAM | JJA | SON | DJF |
|---|---|---|---|---|
| CESM1 | 0.0314 (0.0956, 0.0083) | 0.0286 (0.0774, 0.0111) | 0.0184 (0.0553, 0.0051) | 0.0156 (0.0445, 0.0052) |
| CESM2 | 0.0253 (0.0722, 0.0083) | 0.0208 (0.0534, 0.0090) | 0.0218 (0.0571, 0.0090) | 0.0186 (0.0464, 0.0085) |
| E3SMv1 | 0.0293 (0.0808, 0.0106) | 0.0281 (0.0713, 0.0125) | 0.0194 (0.0529, 0.0073) | 0.0162 (0.0420, 0.0069) |
| MERRA-2 | 0.0465 (0.1095, 0.0236) | 0.0369 (0.0853, 0.0196) | 0.0232 (0.0559, 0.0113) | 0.0221 (0.0501, 0.0119) |
| CALIOP L15 | 0.0332 (0.0799, 0.0170) | 0.0339 (0.0765, 0.0192) | 0.0183 (0.0460, 0.0087) | 0.0173 (0.0407, 0.0092) |
| CALIOP Y15 | 0.0385 (0.0864, 0.0217) | 0.0366 (0.0769, 0.0222) | 0.0248 (0.0523, 0.0150) | 0.0231 (0.0437, 0.0160) |
| MODIS Terra | 0.0788 (0.1333, 0.0595) | 0.0780 (0.1269, 0.0615) | 0.0623 (0.0937, 0.0511) | 0.0607 (0.0953, 0.0504) |
| MODIS Aqua | 0.0706 (0.1209, 0.0529) | 0.0707 (0.1144, 0.0560) | 0.0522 (0.0813, 0.0419) | 0.0569 (0.0918, 0.0464) |
| MISR | (     , 0.0413) | (     , 0.0406) | (     , 0.0351) | (     , 0.0328) |

Note: the values in parentheses are for land and ocean, respectively.















**Figures**

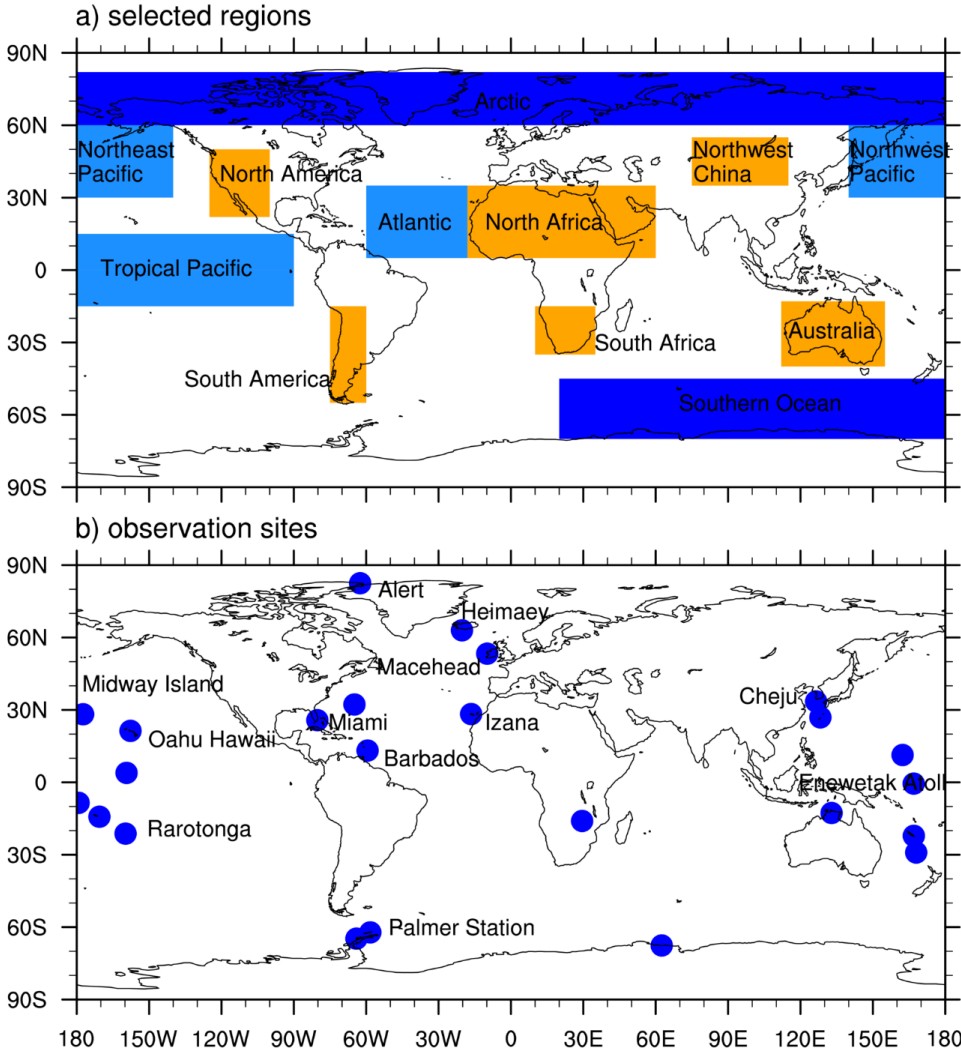

**Figure 1.** Illustration of (a) 12 selected domains and (b) network of stations measuring dust surface
concentrations.








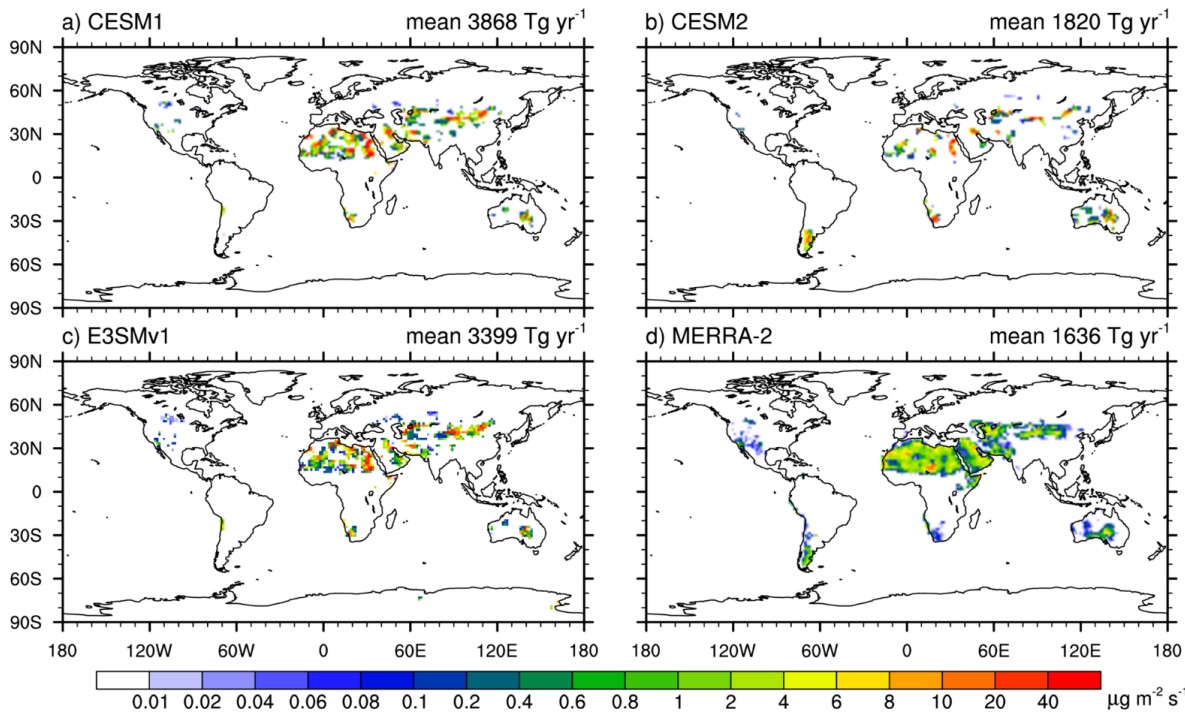

**Figure 2.** Spatial distributions of global annual mean dust emission ($\mu g\ m^{-2}\ s^{-1}$) from model experiments.

The values are global annual mean dust emission.





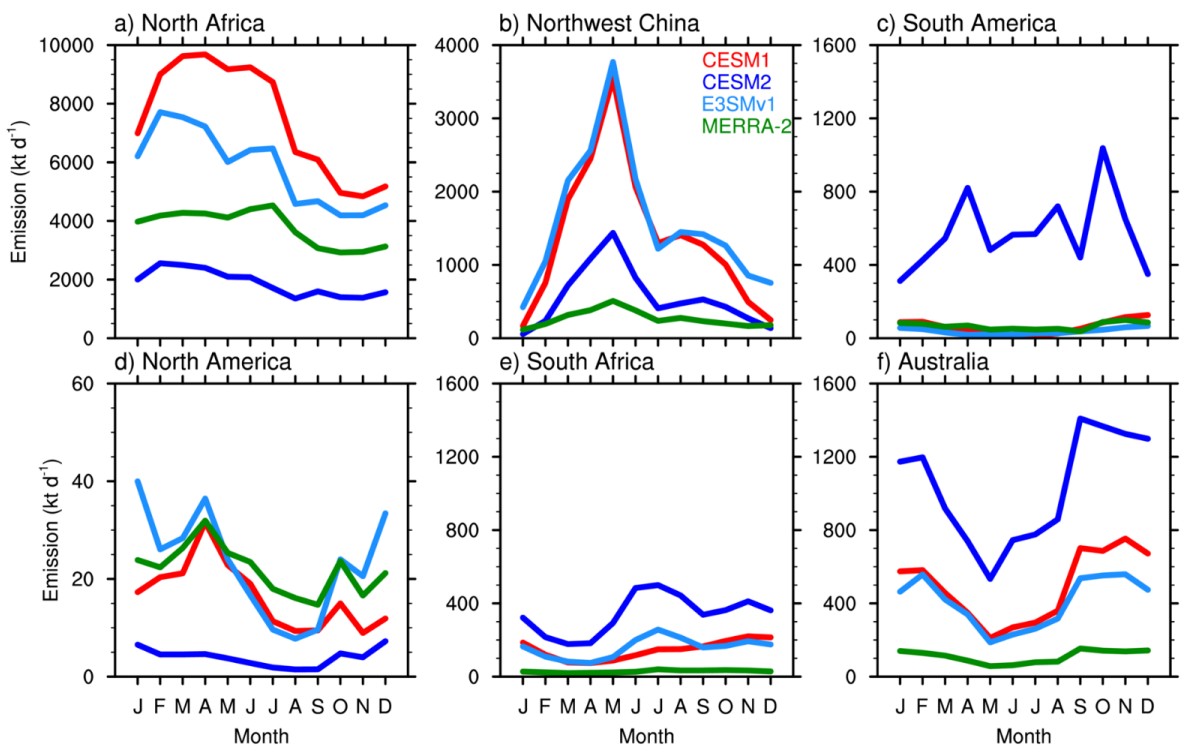


**Figure 3.** Seasonal variations of dust emission (kt d$^{-1}$) in source regions: (a) North Africa, (b) Northwest


China, (c) South America, (d) North America, (e) South Africa, and (f) Australia.
















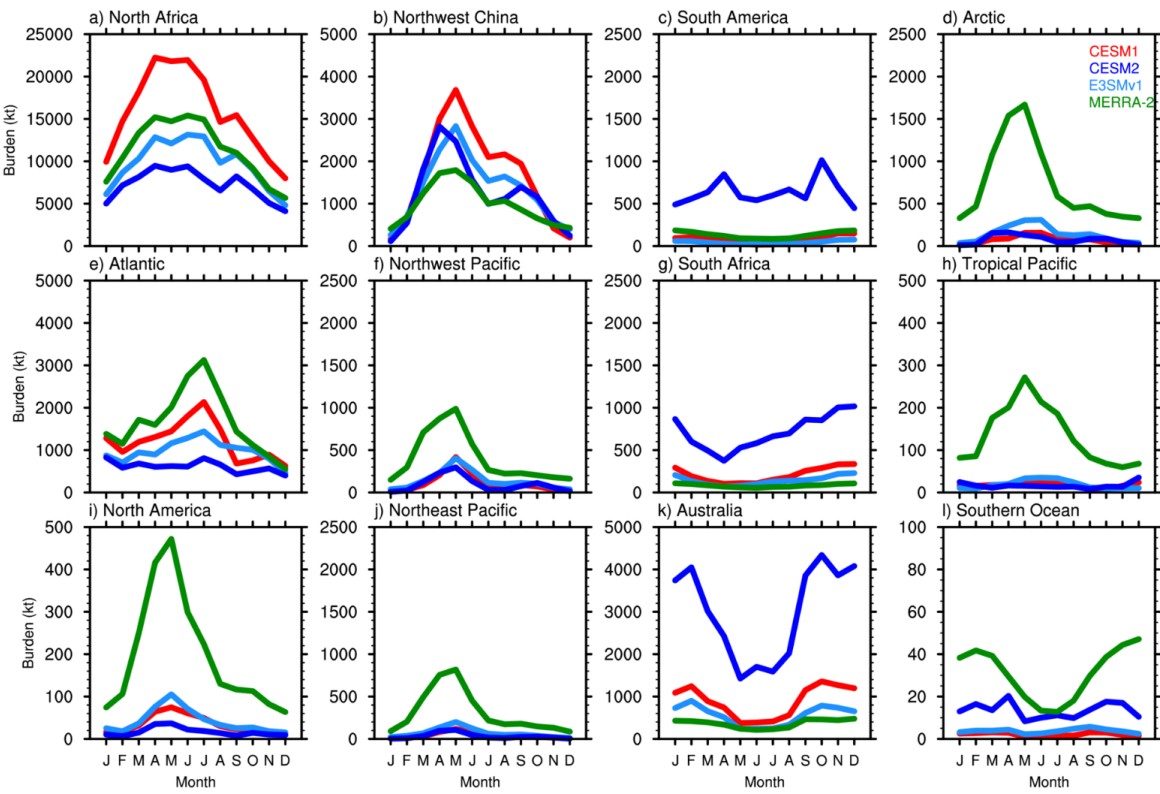


**Figure 4.** Seasonal variations of dust burden (kt) from model experiments over 12 selected regions during

2007-2009.














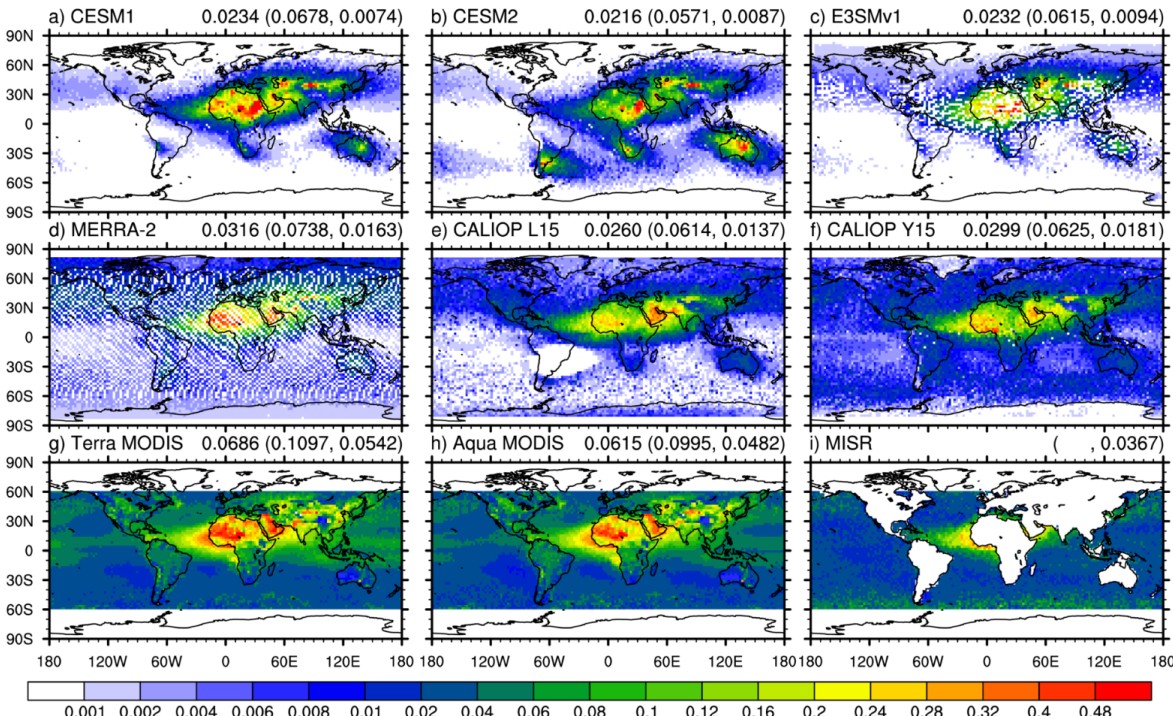

**Figure 5.** Spatial distributions of global annual mean DOD from model experiments, CALIOP, MODIS, and

MISR during 2007-2009. The values are annual mean DOD between 60°S and 60°N. The values in the

parentheses are annual mean DOD over land and ocean, respectively. The stripe pattern of white space in (c)

and (d) is due to the date collocation.





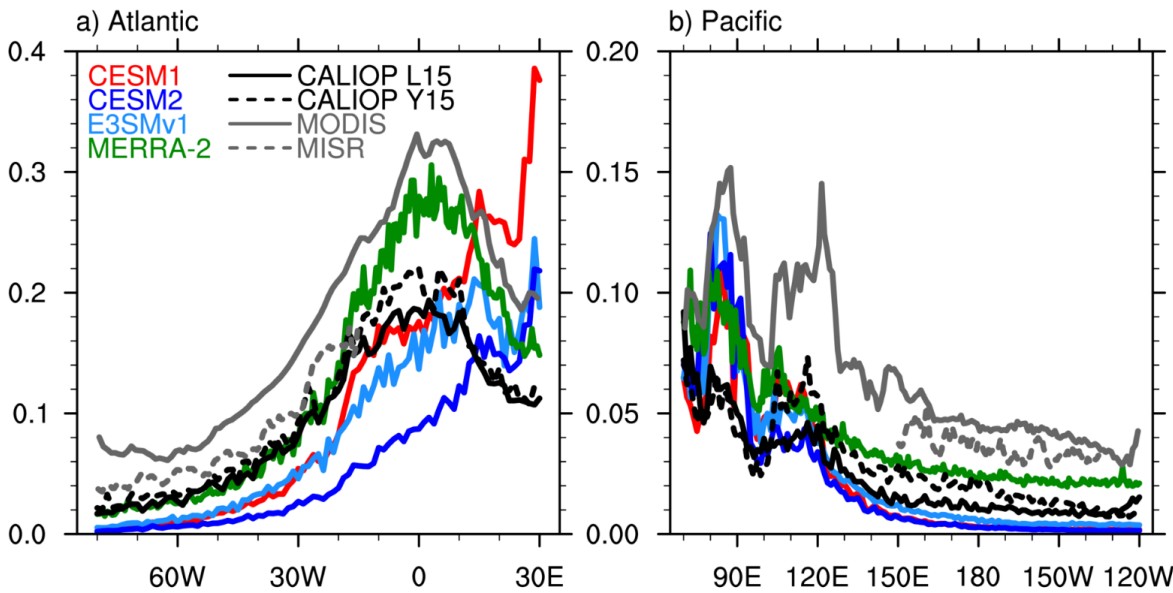

**Figure 6.** Meridional mean of DOD from model experiments, CALIOP, MODIS, and MISR across the (a)

Atlantic (0°-35°N) and (b) Pacific (30°N-60°N).





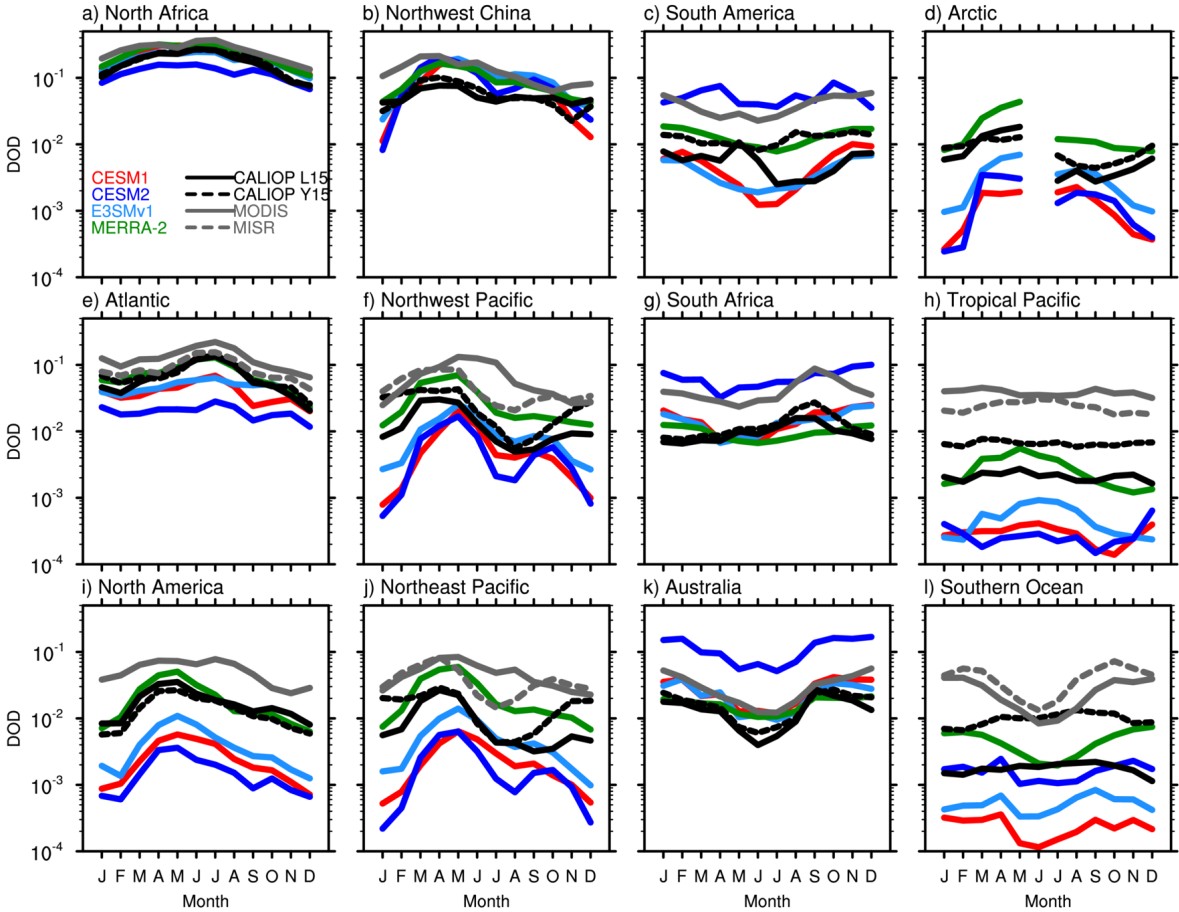

**Figure 7.** Seasonal variations of DOD from model experiments, CALIOP, MODIS, and MISR over 12 selected regions during 2007-2009. The gap in (d) is due to the missing of nighttime data during the polar day.





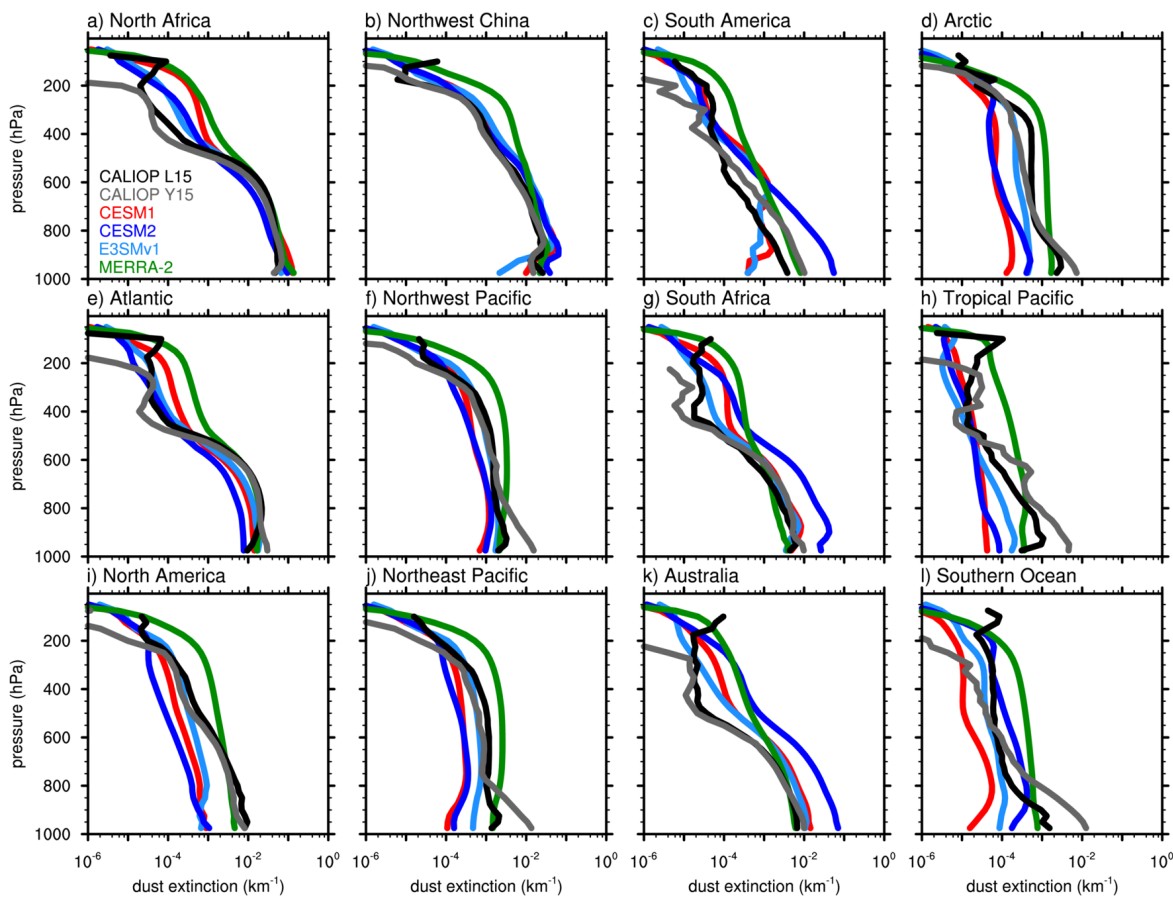

**Figure 8.** Vertical profiles of annual mean dust extinction (km$^{-1}$) from model simulations and CALIOP over

12 selected regions during 2007-2009.



![Figure 9 scatter plots]

**Figure 9.** Observed and simulated annual mean dust surface concentrations (µg m$^{-3}$) at 24 sites. The

measurements at Alert are from Fan (2013); the observations at Heimaey, Barbados, and Miami are from

Prospero et al. (2012); the dataset for the other 20 sites are from Huneeus et al. (2011). These sites were

operated by the University of Miami (Arimoto et al., 1996; Prospero et al., 1989, 1996). Different color

represents different regions.





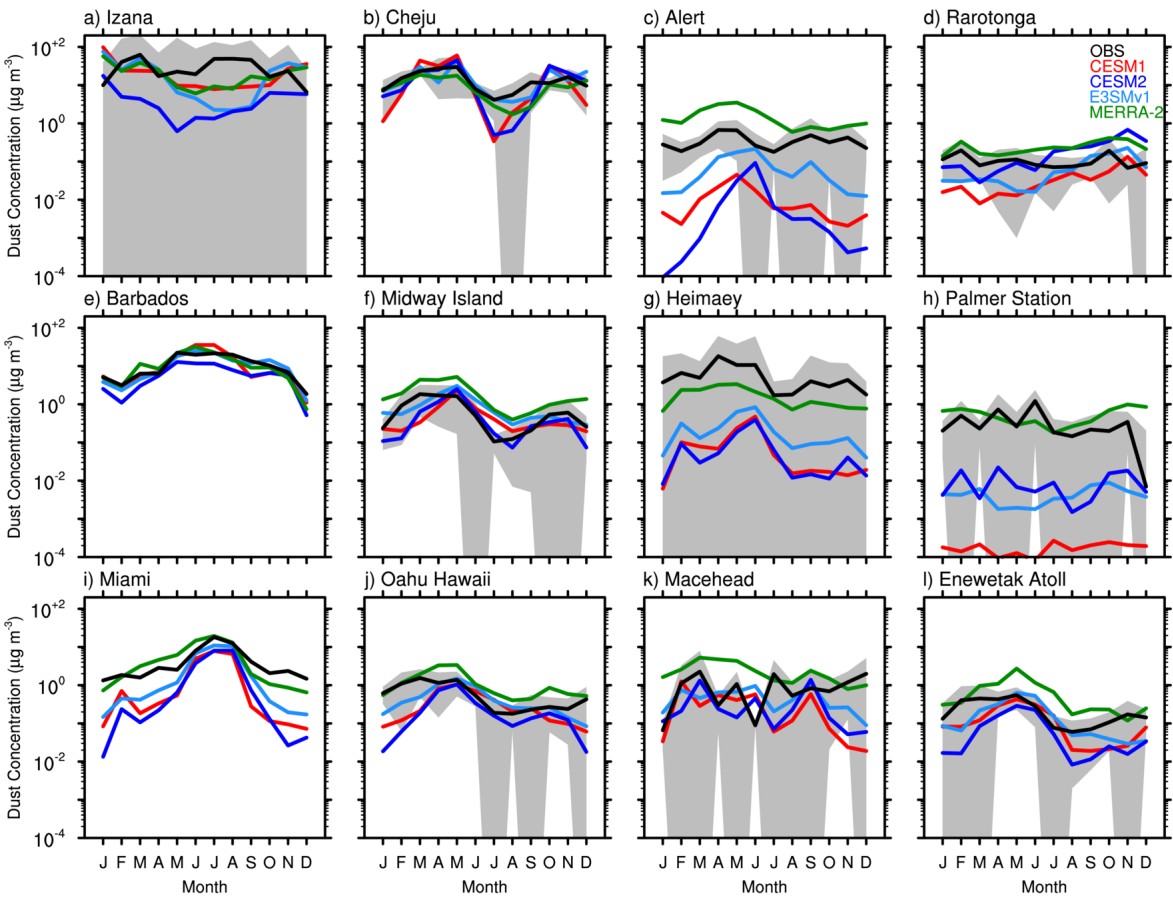

.016

**Figure 10.** Seasonal variations of dust surface concentrations (µg m⁻³) from model simulations and ground

measurements at 12 selected sites. Shaded areas are for plus/minus one standard deviation of observations.

.019

.020

.021

.022

.023

.024

.025