# Peer review of "Understanding Processes that Control Dust Spatial Distributions with Global Climate Models and Satellite Observations"

_Atmospheric Chemistry and Physics, 2020_

## Referee Comment (RC1) · Anonymous Referee #2 · 21 May 2020

General Comments:

Dust particles play important roles in the climate system and local environment, so it's critical to advance the current understanding of how the spatial distribution of dust and relevant processes are represented in the climate models. Here dust mass budget, extinction profile, and surface concentrations from three GCMs (CESM1, CESM2, and E3SMv1) and one reanalysis product (MERRA2) are compared with multiple satellite products, e.g., MODIS, MISR, CALIOP, and station observations. All the models underestimate dust transport over the oceans, although E3SMv1 performs slightly better due to its higher mass faction of fine mode dust. MERRA2 also shows better agreement

with CALIOP DOD. The discrepancies among the satellite products are also discussed.

The paper is overall well written. The authors did a credible job in analyzing how different model settings, such as dust source function, geometric standard deviations, mass fraction, and model layers, affect the simulation of dust in the three GCMs that used the same dust emission scheme. The findings help better understand the performance of the widely used GCMs. I have some comments regarding the methodology, and some clarification probably would further improve the paper.

1. Here model performances are evaluated by comparing model results with satellite retrievals. As noted by the authors, due to the differences in the instruments and retrieval algorithms, certain discrepancies are found among satellite products, adding difficulties to the evaluation. I wonder if the authors can include some discussion on the uncertainties of the satellite products themselves, e.g., their estimated errors in AOD in comparison with ground observations, which probably could be found in previous publications.

2. Some details about model settings are not clear, which may affect the interpretation of model results. For instance, are surface winds nudged in all the GCMs or only in the E3SMv1? If it's only nudged in E3SMv1, how would this affect the intercomparison? All the three models used the DEAD dust emission scheme, is the same tuning factor applied? If not, it is expected to have quite different emissions regardless of other settings.

3. While many factors, such as dust source map and mass faction, can affect dust transport, meteorological conditions may also play a role. I think adding a brief discussion about how meteorological factors could affect dust transport in the models in section 4 would complement current analysis.

Specific Comments:

1. L85, what does "L1B" standard for?

2. L100, It's actually "three models", since MERRA2 is a reanalysis.

3. L129, do you have any idea why the source function in CESM2.1 is so dramatically different from CESM1.2?

4. L143 "fine dust" instead of "find dust"?

5. L164, "see Figure 3 in Kok 2011", wrong citation? I did not find information about MERRA2 in Kok (2011).

6. L168, model levels are inconsistent with the values in Table 1.

7. L183-188, is Ångström exponent <0.3 applied by Pu and Ginoux (2016) or only in the DOD you retrieved?

8. L459-460, can smaller fractions of fine dust in the models also contribute to the biases?

9. L494, what criteria did you use to select the 12 sites? Availability of records? Geographic location?

10. L514, "E3SMv1 produces small amount of dust emission in the Antarctic (Fig. 2c)". It is interesting that CESM1 did not show any dust emission in the Antarctic despite that it used the same source function as E3SMv1. Is this due to different snow coverage?

11. Fig. 1b, station names are labeled for some sites but not others. Why?

12. Fig. 5, is the collocation method similar to that described for dust extinction in L232-241?

---

## Editor Comment (EC1) · Ashu Dastoor (Editor) · 2 Sep 2020

This study provides a comprehensive evaluation of the spatiotemporal variations of dust extinction profiles and dust optical depth simulated by several GCMs against satellite retrievals from Cloud-Aerosol Lidar with Orthogonal Polarization (CALIOP), Moderate Resolution Imaging Spectroradiometer (MODIS), and Multi-angle Imaging SpectroRadiometer (MISR). The study provides a quantitative analysis of the importance of representing dust emission, deposition processes, and size distribution in GCMs for capturing observed dust spatiotemporal distributions. The study also discusses discrepancies among the satellite products.

[Figure]

Dust particles play important roles in the climate, and its understating and accurate simulation is important to advancing climate models and their predictions. The authors have presented an excellent analysis of this topic, helpful to the climate modelers. The manuscript is well written and results are clearly presented. The study is a valuable contribution to advancing modeling of dust in climate models.

―――――――――――――――――

---

## Author Comment (AC1) · 26 Sep 2020

We thank the editor and the anonymous reviewer for the encouraging comments and constructive suggestions on the manuscript. Below, we explain how the comments and suggestions are addressed and make note of the revisions in the revised manuscript. The reviewer's comments are in blue color. Our replies are in black, and our corresponding revisions in the manuscript are in red.

**Editor Comments**

This study provides a comprehensive evaluation of the spatiotemporal variations of dust extinction profiles and dust optical depth simulated by several GCMs against satellite retrievals from Cloud-Aerosol Lidar with Orthogonal Polarization (CALIOP), Moderate Resolution Imaging Spectroradiometer (MODIS), and Multi-angle Imaging SpectroRadiometer (MISR). The study provides a quantitative analysis of the importance of representing dust emission, deposition processes, and size distribution in GCMs for capturing observed dust spatiotemporal distributions. The study also discusses discrepancies among the satellite products.

Dust particles play important roles in the climate, and its understanding and accurate simulation is important to advancing climate models and their predictions. The authors have presented an excellent analysis of this topic, helpful to the climate modelers. The manuscript is well written and results are clearly presented. The study is a valuable contribution to advancing modeling of dust in climate models.

**Reply:** We thank the editor for the encouraging comments. We revised the manuscript according to the anonymous reviewer's comments and suggestions.

---

## Author Comment (AC2) · 26 Sep 2020

We thank the editor and the anonymous reviewer for the encouraging comments and constructive suggestions on the manuscript. Below, we explain how the comments and suggestions are addressed and make note of the revisions in the revised manuscript. The reviewer's comments are in blue color. Our replies are in black, and our corresponding revisions in the manuscript are in red.

**Reviewer #1**

General Comments:
Dust particles play important roles in the climate system and local environment, so it's critical to advance the current understanding of how the spatial distribution of dust and relevant processes are represented in the climate models. Here dust mass budget, extinction profile, and surface concentrations from three GCMs (CESM1, CESM2, and E3SMv1) and one reanalysis product (MERRA2) are compared with multiple satellite products, e.g., MODIS, MISR, CALIOP, and station observations. All the models underestimate dust transport over the oceans, although E3SMv1 performs slightly better due to its higher mass fraction of fine mode dust. MERRA2 also shows better agreement with CALIOP DOD. The discrepancies among the satellite products are also discussed.

The paper is overall well written. The authors did a credible job in analyzing how different model settings, such as dust source function, geometric standard deviations, mass fraction, and model layers, affect the simulation of dust in the three GCMs that used the same dust emission scheme. The findings help better understand the performance of the widely used GCMs. I have some comments regarding the methodology, and some clarification probably would further improve the paper.

**Reply:** We thank the reviewer for the encouraging comments. We have revised the manuscript following your comments regarding the methodology and clarifications of the text to improve the paper.

1. Here model performances are evaluated by comparing model results with satellite retrievals. As noted by the authors, due to the differences in the instruments and retrieval algorithms, certain discrepancies are found among satellite products, adding difficulties to the evaluation. I wonder if the authors can include some discussion on the uncertainties of the satellite products themselves, e.g., their estimated errors in AOD in comparison with ground observations, which probably could be found in previous publications.

**Reply:** We thank the reviewer for the suggestion. We added some discussion on the comparison of AOD retrieved from CALIOP, MODIS and MISR with AERONET ground observations in section 3.2 of the revised manuscript:

"Previous studies found that MODIS and MISR AOD agrees reasonably well with AERONET (e.g., Sayer et al., 2014; Garay et al., 2020), while CALIOP AOD has a notable low bias (e.g., Schuster et al., 2012; Omar et al., 2013; Kim et al., 2018). Sayer et al. (2014) evaluated C6 DB, DT and merged AOD from MODIS Aqua against AERONET observations at 111 sites during 2006-2008. A small median bias of -0.0047 for merged AOD was found if the three products are validated independently. Garay et al. (2020) showed that MISR level 2 V23 AOD has a low bias of -0.002 compared with AERONET observations. Schuster et al. (2012) compared CAL-L2 version 3 AOD with measurements at 147 AERONET sites from June 2006 to May 2009. They found that CALIOP AOD has relative and absolute biases of -13% and -0.029, which is mainly caused by low biases for columns that contain dust subtype. This indicates that a higher lidar ratio (> 40 sr) may be needed to improve CALIPSO dust retrievals."

"More recently, Kim et al. (2018) evaluated CAL-L2 version 3 and 4.10 AOD against measurements from 176 AERONET sites and MODIS level 2 C6 products from 2007 to 2009. They found that global annual mean CAL-L2 AOD has increased from 0.084 in version 3 to 0.128 in version 4.10 for nighttime, which is mostly due to lidar ratio revisions for different aerosol subtypes. The low AOD bias relative to AEROENT is improved from -0.064 in version 3 to -0.051 in version 4.10."

We also added some discussion on the comparison of aerosol extinction retrieved from CALIOP with ground lidar observations in section 4 of the revised manuscript:

"A low bias of the CALIOP aerosol extinction in the lower troposphere (< 2 km) relative to ground-based lidar measurements from the Micro-Pulse Lidar Network (MPLNET) and the European Aerosol Research Lidar Network (EARLINET) at several individual sites has been found in previous studies (e.g., Campbell et al., 2012; Misra et al., 2012; Papagiannopoulos et al., 2016). Further work can be done to evaluate CALIOP dust extinction against measurements from MPLNET and EARLINET."

2. Some details about model settings are not clear, which may affect the interpretation of model results. For instance, are surface winds nudged in all the GCMs or only in the E3SMv1? If it's only nudged in E3SMv1, how would this affect the intercomparison? All the three models used the DEAD dust emission scheme, is the same tuning factor applied? If not, it is expected to have quite different emissions regardless of other settings.

**Reply:** In this study, horizontal wind components u and v at all vertical layers in all three GCMs were nudged toward MERRA-2 meteorology. Since we tuned the global annual mean dust emission in the three GCMs so that AOD in the dust source regions

(DOD/AOD>0.5) matches the satellite observations, different tuning factors were applied. However, the dust emission is changed uniformly over the globe by using a single tuning factor. The spatial distributions of dust emission can still be influenced by other parameter settings, such as source function and soil moisture. We control AOD over source regions so that we can compare the performance of CESM1, CESM2, and E3SMv1 in simulating dust transport from source regions. CESM1 and E3SMv1 produce quite similar dust emission. However, dust emission in CESM2 is much lower due to its longer dust lifetime in the atmosphere to have a similar global mean DOD.

To avoid the confusion, we modified the text in the revised manuscript:

"The horizontal wind components u and v in the three models were all nudged toward the MERRA-2 meteorology using a relaxation time scale of 6 hours."

"The global annual mean dust emission in CESM1.2, CESM2.1, and E3SMv1 was tuned so that AOD in the dusty regions (DOD/AOD>0.5) matches the observations from MODIS onboard Terra and Aqua. Thus, the tuning factors are different among the three models. Generally, CESM1 and E3SMv1 produce quite similar dust emission. However, dust emission in CESM2 is much lower due to its longer dust lifetime in the atmosphere to have a similar global mean DOD."

3. While many factors, such as dust source map and mass fraction, can affect dust transport, meteorological conditions may also play a role. I think adding a brief discussion about how meteorological factors could affect dust transport in the models in section 4 would complement current analysis.

**Reply:** We thank the reviewer for the great suggestion. We added a brief discussion on the effects of meteorological factors on dust transport in section 4 of the revised manuscript:

"Smith et al. (2017) ran CAM4 with constrained meteorology (i.e., horizontal wind components, temperature, surface pressure, sensible and latent heat fluxes, and wind stress) from three reanalysis (MERRA, ERA-interim, and NCEP) and found that the global annual mean AOD is 0.026 ± 30%, indicating an uncertainty due to meteorology of 30%. Precipitation is another important meteorological factor which not only affects the dust transport by wet deposition but also changes dust emission through soil moisture. A high bias of precipitation over and near the source regions may reduce dust transport to remote regions. Rasch et al. (2019) showed that E3SMv1 and CESM1 tend to rain too early compared with observations, especially over land (~ 6 hours). The bias in the diurnal cycle of precipitation may also influence the dust transport, considering the strong

vertical mixing of dust during daytime."

**Specific Comments:**
1. L85, what does "L1B" standard for?

**Reply:** It stands for level 1B. We changed "CAL-L1B" to "CALIOP level 1B (CAL-L1B)" in the revised manuscript.

2. L100, it's actually "three models", since MERRA2 is a reanalysis.

**Reply:** We changed "four models" to "three models and one reanalysis" in the revised manuscript.

3. L129, do you have any idea why the source function in CESM2.1 is so dramatically different from CESM1.2.

**Reply:** We can see from Fig. S1 that the source function in CESM2.1 is tuned according to different regions. We contacted Dr. Natalie Mahowald from Cornell University and know that the source function in CESM2.1 was tuned down to match the observed global DOD because of the large differences in aerosol coarse mode size and standard deviation between CESM1 and CESM2.

4. L143 "fine dust" instead of "find dust"?

**Reply:** Corrected. Thanks.

5. L164, "see Figure 3 in Kok 2011", wrong citation? I did not find information about MERRA2 in Kok (2011).

**Reply:** In MERRA-2, the size distribution of emitted dust particles follows Tegen and Lacis (1996). Figure 3 in Kok 2011 shows the emitted dust size distributions for Tegen and Lacis (1996) (magenta lines) and Zender et al. (2003) (green lines). We modified the sentence to clarify that:

"MERRA-2 uses the emitted dust size distribution following Tegen and Lacis (1996) and has the highest mass fraction of emitted fine dust (0.1-1.0 μm) among the three models and one reanalysis (see Figure 3 in Kok 2011 for the comparison of emitted dust size distribution), which can increase the dust transport."

6. Line 168, model levels are inconsistent with the values in Table 1.

**Reply:** We changed the vertical levels to 56 for CESM1 and CESM2 in Table1. Thanks.

7. L183-188, is Ångström exponent <0.3 applied by Pu and Ginoux (2016) or only in the DOD you retrieved?

**Reply:** Ångström exponent <0.3 is applied by Pu and Ginoux (2016). We use the MODIS DOD over land provided by Dr. Paul Ginoux.

8. L459-460, can smaller fractions of fine dust in the models also contribute to the biases?

**Reply:** Yes, smaller fractions of fine dust in the models can be a contributing factor to the low biases. However, Adebiyi et al. (2020) found that current GCMs overestimate the amount of fine dust (diameter less than 2.5 μm) in the atmosphere when compared to measurements. It would be nice to have size distribution measurements of dust over the Pacific to investigate this possible issue.

9. Lines 494, what criteria did you use to select the 12 sites? Availability of records? Geographic location?

**Reply:** We selected the 12 sites mainly based on their geographic locations, which cover the Arctic, Antarctic, trans-Pacific region, and trans-Atlantic region. We added one sentence in the revised manuscript to explain that:

"We select the 12 sites based on their geographic locations, which cover the Arctic, Antarctic, trans-Pacific region, and trans-Atlantic region."

10. L514, "E3SMv1 produces small amount of dust emission in the Arctic (Fig. 2c)". It is interesting that CESM1 did not show any dust emission in the Antarctic despite that it is

used the same source functions as E3SMv1. Is this due to different snow coverage?

**Reply:** We think it is mainly due to low soil moisture along the coast of the Antarctic in E3SMv1. We added a sentence to clarify that:

"E3SMv1 produces small amount of dust emission in the Antarctic (Fig. 2c) due to its low soil moisture along the coast of the Antarctic."

11. Fig. 1b, station names are labeled for some sites but no others. Why?

**Reply:** We previously only labeled the 12 sites shown in Fig. 10. We revised Fig. 1 to label all observation sites used in Fig. 9.

[Figure]

**Figure 1.** Illustration of (a) 12 selected domains and (b) network of stations measuring dust surface concentrations.

12. Fig. 5, is the collocation method similar to that described for dust extinction in

**Reply:** Yes. We mentioned in Section 2.3.2 that we first collocate modeled dust extinction with CALIOP retrievals and then integrate it to get the DOD values. Note that DOD from model and CALIOP is for nighttime, while DOD from MODIS and MISR is for daytime. We added a note in the figure caption:

"We integrate the collocated dust extinction profiles from the three models and one reanalysis to get the nighttime DOD values. DOD from MODIS and MISR is for daytime."

**References**

Adebiyi, A. A., Kok, J. F., Wang, Y., Ito, A., Ridley, D. A., Nabat, P., and Zhao, C.: Dust Constraints from joint Observational-Modelling-experiMental analysis (DustCOMM): comparison with measurements and model simulations, Atmos. Chem. Phys., 20, 829–863, https://doi.org/10.5194/acp-20-829-2020, 2020.

Campbell, J. R., Tackett, J. L., Reid, J. S., Zhang, J., Curtis, C. A., Hyer, E. J., Sessions, W. R., Westphal, D. L., Prospero, J. M., Welton, E. J., Omar, A. H., Vaughan, M. A., and Winker, D. M.: Evaluating nighttime CALIOP 0.532 µm aerosol optical depth and extinction coefficient retrievals, Atmos. Meas. Tech., 5, 2143–2160, https://doi.org/10.5194/amt-5-2143-2012, 2012.

Garay, M. J., Witek, M. L., Kahn, R. A., Seidel, F. C., Limbacher, J. A., Bull, M. A., Diner, D. J., Hansen, E. G., Kalashnikova, O. V., Lee, H., Nastan, A. M., and Yu, Y.: Introducing the 4.4 km spatial resolution Multi-Angle Imaging SpectroRadiometer (MISR) aerosol product, Atmos. Meas. Tech., 13, 593–628, https://doi.org/10.5194/amt-13-593-2020, 2020.

Kim, M.-H., Omar, A. H., Tackett, J. L., Vaughan, M. A., Winker, D. M., Trepte, C. R., Hu, Y., Liu, Z., Poole, L. R., Pitts, M. C., Kar, J., and Magill, B. E.: The CALIPSO version 4 automated aerosol classification and lidar ratio selection algorithm, Atmos.

Meas. Tech., 11, 6107–6135, https://doi.org/10.5194/amt-11-6107-2018, 2018.

Kok, J. F.: A scaling theory for the size distribution of emitted dust aerosols suggests climate models underestimate the size of the global dust cycle, P. Natl. Acad. Sci. USA., 108, 1016-1021, https://doi.org/10.1073/pnas.1014798108, 2011.

Misra, A., Tripathi, S. N., Kaul, D. S., and Welton, E. J.: Study of MPLNET-derived aerosol climatology over Kanpur, India, and validation of CALIPSO level 2 version 3 backscatter and extinction products, J. Atmos. Ocean. Tech., 29, 1285-1294, https://doi.org/10.1175/JTECH-D-11-00162.1, 2012.

Omar, A. H., Winker, D. M., Tackett, J. L., Giles, D. M., Kar, J., Liu, Z., Vaughan, M. A., Powell, K. A., Trepte, C. R.: CALIOP and AERONET aerosol optical depth comparisons: One size fits one, J. Geophys. Atmos., 118, 4748-4766, https://doi.org/10.1002/jgrd.50330, 2013.

Papagiannopoulos, N., Mona, L., Alados-Arboledas, L., Amiridis, V., Baars, H., Binietoglou, I., Bortoli, D., D'Amico, G., Giunta, A., Guerrero-Rascado, J. L., Schwarz, A., Pereira, S., Spinelli, N., Wandinger, U., Wang, X., and Pappalardo, G.: CALIPSO climatological products: evaluation and suggestions from EARLINET, Atmos. Chem. Phys., 16, 2341–2357, https://doi.org/10.5194/acp-16-2341-2016, 2016.

Sayer, A. M., Munchak, L. A., Hsu, N. C., Levy, R. C., Bettenhausen, C., and Jeong, M.-J.: MODIS Collection 6 aerosol products: Comparison between Aqua's e-Deep Blue, Dark Target, and "merged" data sets, and usage recommendations, J. Geophys. Res.-Atmos., 119, 13965-13989, https://doi.org/10.1002/2014JD022453, 2014.

Schuster, G. L., Vaughan, M., MacDonnell, D., Su, W., Winker, D., Dubovik, O., Lapyonok, T., and Trepte, C.: Comparison of CALIPSO aerosol optical depth retrievals to AERONET measurements, and a climatology for the lidar ratio of dust, Atmos. Chem. Phys., 12, 7431–7452, https://doi.org/10.5194/acp-12-7431-2012, 2012.

Smith, M. B., Mahowald, N. M., Albani, S., Perry, A., Losno, R., Qu, Z., Marticorena, B., Ridley, D. A., and Heald, C. L.: Sensitivity of the interannual variability of mineral aerosol simulations to meteorological forcing dataset, Atmos. Chem. Phys., 17, 3253–3278, https://doi.org/10.5194/acp-17-3253-2017, 2017.